# TRANSIC: Sim-to-Real Policy Transfer by Learning from Online Correction

**Yunfan Jiang, Chen Wang, Ruohan Zhang, Jiajun Wu, Li Fei-Fei**

Stanford University

**Abstract:** Learning in simulation and transferring the learned policy to the real world has the potential to enable generalist robots. The key challenge of this approach is to address simulation-to-reality (sim-to-real) gaps. Previous methods often require domain-specific knowledge *a priori*. We argue that a straightforward way to obtain such knowledge is by asking humans to observe and assist robot policy execution in the real world. The robots can then learn from humans to close various sim-to-real gaps. We propose TRANSIC, a data-driven approach to enable successful sim-to-real transfer based on a human-in-the-loop framework. TRANSIC allows humans to augment simulation policies to overcome various unmodeled sim-to-real gaps holistically through intervention and online correction. Residual policies can be learned from human corrections and integrated with simulation policies for autonomous execution. We show that our approach can achieve successful sim-to-real transfer in complex and contact-rich manipulation tasks such as furniture assembly. Through synergistic integration of policies learned in simulation and from humans, TRANSIC is effective as a holistic approach to addressing various, often coexisting sim-to-real gaps. It displays attractive properties such as scaling with human effort. Videos and code are available at `transic-robot.github.io`.

**Keywords:** Sim-to-Real Transfer, Human-in-the-Loop, Human Correction

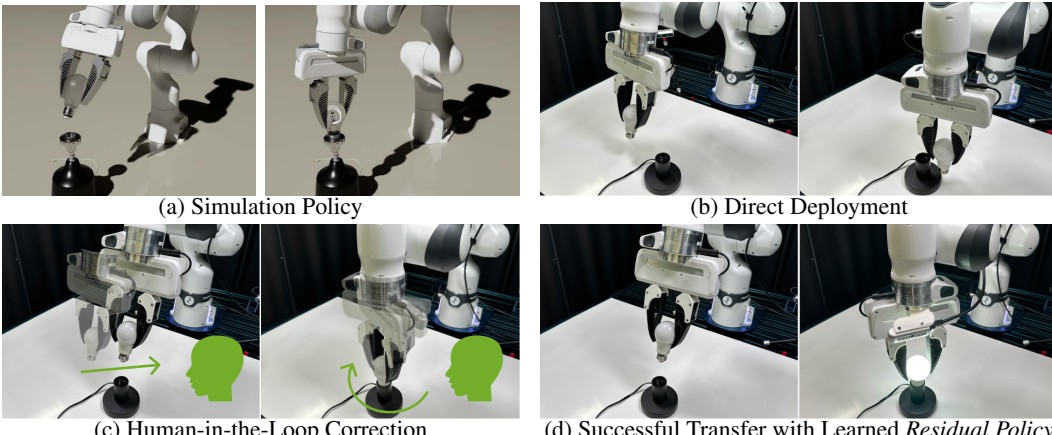

(a) Simulation Policy          (b) Direct Deployment

(c) Human-in-the-Loop Correction      (d) Successful Transfer with Learned *Residual Policy*

Figure 1: **TRANSIC for sim-to-real transfer in contact-rich robotic manipulation tasks. a) and b)** Naïvely deploying policies trained in simulation usually fails due to various sim-to-real gaps. Here, the robot attempts to first align the light bulb with the base and then insert and screw the light bulb into the base. **c)** A human operator monitors robot behaviors, intervenes, and provides online correction through teleoperation when necessary. Human data are collected to train a *residual policy* to tackle various sim-to-real gaps in a holistic manner. **d)** The simulation and the residual policies are integrated together during test time to achieve a successful sim-to-real transfer for contact-rich tasks, such as screwing a light bulb into the base.

8th Conference on Robot Learning (CoRL 2024), Munich, Germany.

# 1 Introduction

Learning in simulation is a potential approach to the realization of generalist robots capable of solving sophisticated decision-making tasks [1, 2]. Learning to solve these tasks requires a large amount of training data [3–5]. Providing unlimited training supervision through state-of-the-art simulation [6–10] could alleviate the burden of collecting data in the real world with physical robots [11, 12]. Therefore, it is crucial to seamlessly transfer and deploy robot control policies acquired in simulation, usually through reinforcement learning (RL), to real-world hardware. Successful demonstrations of this simulation-to-reality (sim-to-real) approach have been shown in dexterous in-hand manipulation [13–17], locomotion [18–27], and quadrotor flight [28, 29].

Nevertheless, replicating similar success in manipulation tasks with robotic arms remains surprisingly challenging, with only a few cases in simple non-prehensile manipulation [30–33], industry assembly under restricted settings [34–38], and peg swinging [39]. The difficulty mainly stems from the unavoidable sim-to-real gaps [40], including but not limited to perception gap [41–43], embodiment mismatch [18, 44, 45], controller inaccuracy [46–48], and dynamics realism [49]. Traditionally, researchers tackle them through system identification [18, 30, 50, 51], domain randomization [13, 52–55], real-world adaptation [56, 57], and simulator augmentation [58–60]. Many of these approaches require explicit, domain-specific knowledge and expertise in tasks or simulators. Although for a particular simulation-reality pair, there may exist specific inductive biases that can be hand-crafted *post hoc* to close the sim-to-real gap [18], this knowledge is often not available *a priori*. Identifying its effects on task completion is also intractable.

We argue that a straightforward and feasible way for humans to obtain such knowledge is to observe and assist policy execution in the real world. If humans can assist the robot to successfully accomplish the tasks in the real world, sim-to-real gaps are effectively addressed. This naturally leads to a generally applicable paradigm that can cover different priors across simulations and realities— human-in-the-loop learning [61–63] and shared-autonomy [64, 65].

Our key insight is that the human-in-the-loop framework is promising for addressing the sim-to-real gaps as a whole, in which humans directly assist the physical robots during policy execution by providing online correction signals. The knowledge required to close sim-to-real gaps can be learned from human signals. We present TRANSIC (**tran**sferring policies **si**m-to-real by learning from online **c**orrection, Fig. 1), a data-driven approach to enable the successful transfer of robot manipulation policies trained with RL in simulation to the real world. In TRANSIC, once the base robot policies are acquired from simulation training, they are deployed to real robots where human operators monitor the execution. When the robot makes mistakes or gets stuck, humans interrupt and assist robot policies through teleoperation. Such human intervention data are collected to train a *residual policy*, after which the base policy and the residual policy are combined to solve contact-rich manipulation tasks. Unlike previous approaches that heavily rely on domain knowledge, since humans can successfully assist the robot trained in silico to complete real-world tasks, sim-to-real gaps are implicitly handled and addressed by humans in a domain-agnostic manner.

To summarize, our key contribution is a **novel, holistic human-in-the-loop method** called TRANSIC to tackle sim-to-real policy transfer for manipulation tasks. Through extensive evaluation, we show that our method leads to **more effective sim-to-real transfer** compared to traditional methods [50, 52] and **requires less real-robot data** compared to the prevalent imitation learning and offline RL algorithms [66–69]. We demonstrate that successful sim-to-real transfer of short-horizon skills can solve **long-horizon, contact-rich manipulation** tasks, such as furniture assembly.

# 2 Sim-to-Real Policy Transfer by Learning from Online Correction

An overview of TRANSIC is shown in Fig. 2. This section starts with a brief preliminary review, followed by a description of simulation training. We then introduce residual policies learned from human intervention and online correction and present an integrated framework for deployment during testing. Lastly, we provide implementation details.

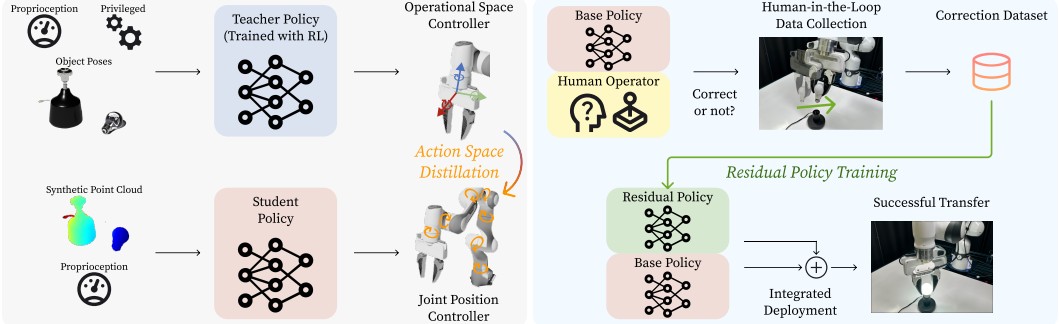

(a) Simulation Policy Training through *Action Space Distillation*    (b) *Residual Policy* Learning from Human Correction

Figure 2: **TRANSIC method overview. a)** Base policies are first trained in simulation through *action space distillation* with demonstrations generated by RL teacher policies. Base policies take point cloud as input to reduce perception gap. **b)** The acquired base policies are first deployed with a human operator monitoring the execution. The human intervenes and corrects through teleoperation when necessary. Such correction data are collected to learn *residual policies*. Finally, both residual policies and base policies are integrated during test time to achieve a successful transfer.

## 2.1 Preliminaries

We formulate a robot manipulation task as an infinite-horizon discrete-time Markov Decision Process (MDP) $\mathcal{M} := (\mathcal{S}, \mathcal{A}, \mathcal{T}, R, \gamma, \rho_0)$, where $\mathcal{S}$ is the state space, and $\mathcal{A}$ is the action space. At time step $t$, a robot observes $s_t \in \mathcal{S}$, executes an action $a_t$, and receives a scalar reward $r_t$ from the reward function $R(s_t, a_t)$. The environment progresses to the next state following the transition function $\mathcal{T}(s_{t+1}|s_t, a_t)$. The robot learns a parameterized policy $\pi_\theta(\cdot|s)$ to maximize the expected discounted return $\mathcal{J} := \mathbb{E}_{\tau \sim p_{\pi_\theta}}[\sum_{t=0}^{\infty} \gamma^t r_t]$ over induced trajectory distribution $\tau := (s_0, a_0, r_0, ...) \sim p_{\pi_\theta}$, where $s_0 \sim \rho_0$ is sampled from the initial state distribution and $\gamma \in [0, 1)$ is a discount factor. We consider simulation and real environments as two different MDPs. We adopt an intervention-based learning framework [66, 67] where a human operator can intervene and take control during the execution of the robot policy.

## 2.2 Learning Base Policies in Simulation with RL

**Policy Learning with 3D Representation**    Object geometry matters for contact-rich manipulation. For example, a robot should ideally insert a light bulb into the lamp base with the thread facing down. To retain such 3D information and facilitate sim-to-real transfer, we propose to use point cloud as the main visual modality. Typical RGB observation used in visuomotor policy training [70] suffers from several drawbacks that hinder successful transfer, such as vulnerability to different camera poses [71] and discrepancies between synthetic and real images [42]. Well-calibrated point cloud observation can bypass these issues and has been successfully demonstrated [14, 72]. During the simulation RL training phase, we synthesize point cloud observations for higher throughput. Concretely, given the synthetic point cloud of the $m$-th object $\mathbf{P}^{(m)} \in \mathbb{R}^{K \times 3}$, we transform it into the global frame through $\mathbf{P}_g^{(m)} = \mathbf{P}^{(m)}(\mathbf{R}^{(m)})^\mathsf{T} + (\mathbf{p}^{(m)})^\mathsf{T}$. Here, $\mathbf{R}^{(m)} \in \mathbb{R}^{3 \times 3}$ and $\mathbf{p}^{(m)} \in \mathbb{R}^{3 \times 1}$ denote the object's orientation and translation in the global frame. Further, the point cloud representation of a scene $\mathbf{S}$ with $M$ objects is aggregated as $\mathbf{P}^{\mathbf{S}} = \bigcup_{m=1}^{M} \mathbf{P}_g^{(m)}$ and subsequently used as policy input.

**Action Space Distillation**    A suitable action abstraction is critical for efficient learning [46, 47] as well as sim-to-real transfer [48]. A high-level controller such as the operational space controller (OSC) [73] facilitates RL exploration [46] but may hinder sim-to-real transfer because it requires accurate modeling of robot parameters, such as joint friction, mass, and inertia [74]; on the other hand, a low-level action space such as the joint position ensures consistent deployment in simulation and real hardware, but renders trial-and-error RL impractical. We draw inspiration from the teacher-student framework [15, 75–77] and propose to first train the teacher policy $\pi^{teacher}$ through RL with OSC and then distill successful trajectories into the student policy $\pi^{student}$ with joint position

control. Specifically, we roll out $\pi^{teacher}$ and record the robot's joint position at every simulated time interval $\delta t$ to construct a dataset $\mathcal{D}^{teacher} = \{\tau^{(n)}\}_{n=1}^N$. We then relabel actions from the end-effector's poses to joint positions. Such a relabeled dataset is ready to train student policies through behavior cloning. We name this approach as *action space distillation* and find it crucial to overcome the sim-to-real controller gap. Furthermore, teacher policies directly receive privileged observations for ease of learning, while student policies learn on synthetic point-cloud inputs to match real-world measurements. The student policy parameterized by $\theta$, $\pi_\theta^{student}$, is trained by minimizing the loss function: $\mathcal{L}^{student} = -\mathbb{E}_{\mathcal{D}^{teacher}} \left[ \log \pi_\theta^{student} \right] + \beta \mathbb{E}_{\mathcal{D}^{pcd}} \left[ \|\phi(\mathbf{P}^{real}) - \phi(\mathbf{P}^{sim})\|^2 \right]$, where $\phi(\cdot)$ denotes the point cloud encoder of $\pi_\theta^{student}$ and $\mathcal{D}^{pcd} = \{(\mathbf{P}^{real}, \mathbf{P}^{sim})^{(i)}\}_{i=1}^N$ is a separate dataset that contains $N$ pairs of matched point clouds in simulation and reality for regularization purpose. Further justifications of this distillation phase can be found in Appendix D.1.

## 2.3 Learning Residual Policies from Online Correction

**Human-in-the-Loop Data Collection** Once the student policy is obtained from simulation, it is used directly as the base policy $\pi^B$ to bootstrap the data collection. Naïvely deploying the base policy on real robots usually results in inferior performance and unsafe motion due to various sim-to-real gaps. In TRANSIC, the base policy is instead deployed in a way that is fully synchronized with the human operator. Concretely, at time step $t$, once $a_t^B \sim \pi^B$ is deployed, a human operator needs to decide whether intervention is necessary, indicated as $\mathbb{1}_t^H$. Intervention is not necessary for most task execution when the robot is approaching objects of interest. However, when the robot tends to behave abnormally, the human operator intervenes and takes full control through teleoperation to correct robot errors. In these cases, the robot's pre- and post-intervention states, as well as intervention indicator $\mathbb{1}_t^H$, are collected to construct the online correction dataset $\mathcal{D}^H \leftarrow \mathcal{D}^H \cup \left( \mathbb{1}_t^H, \mathbf{q}_t^{pre}, \mathbf{q}_t^{post} \right)$. This procedure is illustrated in Appendix Algorithm 1.

**Human Correction as Residual Policies** Properly modeling human correction can be challenging. This is because humans usually solve tasks not purely based on current observation, hence the non-Markovian decision process [68]. Therefore, directly fine-tuning the base policy $\pi^B$ on human correction dataset $\mathcal{D}^H$ leads to large motions and even model collapse (Sec. 3). Inspired by prior work on learning residuals to compensate for unknown dynamics and noisy observations [78–80], we propose to incorporate human correction behaviors with *residual policies*. Concretely, at the time of intervention, a residual policy $\pi_\psi^R$ parameterized by $\psi$ learns to predict human intervention as the difference between post- and pre-intervention robot states: $a^R = \mathbf{q}^{post} \ominus \mathbf{q}^{pre}$, where $\ominus$ denotes generalized subtraction. For continuous variables such as joint position, it computes the numerical difference; for binary variables such as opening and closing the gripper, it computes exclusive nor. The residual policy is then trained to maximize the likelihood of human correction: $\mathcal{L}^{residual} = -\mathbb{E}_{\mathcal{D}^H} \left[ \log \pi_\psi^R(a^R|\cdot) \right]$.

## 2.4 An Integrated Deployment Framework

In practice, we find that learning a gating function to control whether to apply residual actions or not leads to better empirical performance (Appendix D.3). We call this *learned gated residual policy*. Denote the gating function as $g_\psi(\cdot)$. It shares the same feature encoder with the residual policy $\pi^R$ and is jointly learned through classification on the same correction dataset $\mathcal{D}^H$. At the inference time, we effectively use $g_\psi$ as an indicator function $\mathbb{1}_g$ to determine whether to apply the predicted residual actions. The policy effectively being deployed to autonomously complete tasks is an integration of base policy $\pi^B$ and residual policy $\pi^R$, gated by $g$: $\pi^{deployed} = \pi^B \oplus \mathbb{1}_g \pi^R$. A joint position controller is used during deployment.

## 2.5 Implementation Details

We use the Isaac Gym [9] simulator. Proximal policy optimization (PPO) [81] is used to train teacher policies from scratch with task-specific reward functions and curricula. Student policies

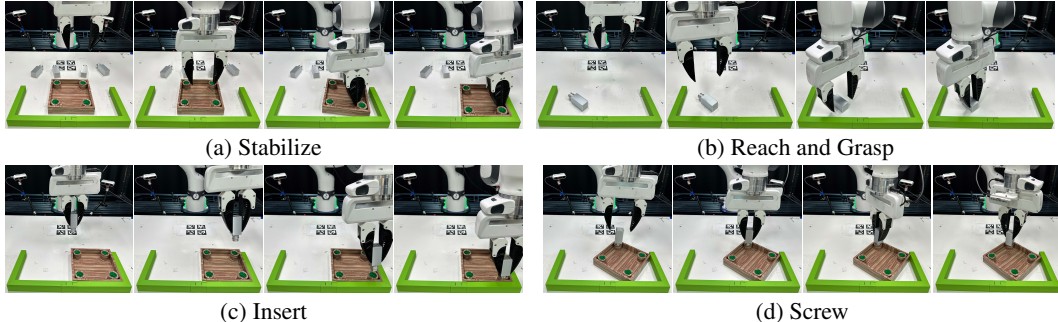

(a) Stabilize                 (b) Reach and Grasp

(c) Insert                   (d) Screw

Figure 3: **Four tasks benchmarked in this work.** They are fundamental skills required to assemble a square table from FurnitureBench [85]. The task definition can be found in Appendix C.1.

are parameterized as Gaussian Mixture Models (GMMs) [68]. See Appendix A for more details. During human-in-the-loop data collection, we use a 3Dconnexion SpaceMouse as the teleoperation interface. Residual policies use state-of-the-art point cloud encoders [82–84] and GMM as the action head. More training hyperparameters are provided in Appendix B.4.

## 3   Experiments

We answer the following research questions through experiments.

$\mathcal{Q}$1: Does TRANSIC lead to better transfer performance while requiring less real-world data?

$\mathcal{Q}$2: How effectively can TRANSIC address different types of sim-to-real gaps?

$\mathcal{Q}$3: How does TRANSIC scale with human effort?

$\mathcal{Q}$4: Does TRANSIC exhibit intriguing properties, such as generalization to unseen objects, policy robustness, ability to solve long-horizon tasks, and other emergent behaviors?

### 3.1   Tasks, Baselines, and Evaluation Protocol

As shown in Fig. 3, we consider complex contact-rich manipulation tasks (*Stabilize*, *Reach and Grasp*, *Insert*, and *Screw*) that require high precision in FurnitureBench [85]. These tasks are challenging and ideal for testing sim-to-real transfer, since perception, embodiment, controller, and dynamics gaps all need to be addressed. We collect 20, 100, 90, and 17 real-robot trajectories with human correction, respectively. These amount to 62, 434, 489, and 58 corrections for each task. See Appendix B.1 for the detailed system setup. We compare with three groups of baselines. **1) Traditional sim-to-real methods:** It includes domain randomization and data augmentation [52] ("DR. & Data Aug."), real-world fine-tuning through BC ("BC Fine-Tune") and implicit Q-learning [69] ("IQL Fine-Tune"). To estimate the performance lower bound, we also include "Direct Transfer" without any data augmentation or real-world fine-tuning. **2) Interactive imitation learning (IL):** It includes HG-Dagger [66] and IWR [67]. **3) Learning from real-robot data only:** It includes BC [86], BC-RNN [68], and IQL [69] that are trained on real-robot demonstrations only. *All* evaluations are conducted on the real robot and consist of 20 trials starting with different objects and robot poses. See Appendix C for details.

### 3.2   Results

**TRANSIC is effective for sim-to-real transfer and requires significantly less real-world data ($\mathcal{Q}$1).** As shown in Fig. 4 and Table A.XI, TRANSIC achieves the best performance on average and in all four tasks with significant margins. What are the reasons for successful transfer? We observe that adding real-world human correction data does not guarantee improvement. For example, among traditional sim-to-real methods, the best baseline BC Fine-Tune outperforms DR. & Data Aug. by 7%, but IQL Fine-Tune leads to worse performance. In contrast, TRANSIC effectively uses human correction data, which boosts average performance by 1.24×. Not only does it achieve

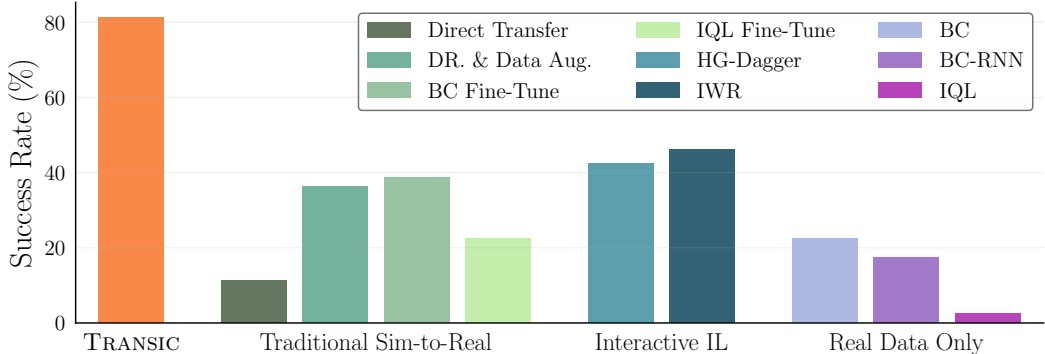

Figure 4: **Average success rates over four benchmarked tasks.** Numerical results in Table A.XI.

the best transfer performance, but it also improves simulation policies the most among various sim-to-real approaches.

Furthermore, TRANSIC outperforms interactive IL methods, including HG-Dagger and IWR, by $0.75\times$ on average. Although both of them weigh the intervention data higher during training, we find that they tend to *erase* the original policy and lead to catastrophic forgetting. In contrast, by incorporating human correction with a separate residual policy and integrating both base and residual policies through gating, TRANSIC combines the best properties of both policies during deployment. It relies on the simulation policy for robust execution most of the time; when the base policy is likely to fail, it automatically applies the residual policy to prevent failures and correct mistakes.

Finally, TRANSIC only requires dozens of real-robot corrections to achieve superior performance. However, methods such as BC-RNN and IQL trained on such a limited number of trajectories suffer from overfitting and model collapse. TRANSIC achieves $3.6\times$ better performance than them. This result highlights the importance of first training in simulation and then leveraging sim-to-real transfer for robot learning practitioners.

**In summary**, we show that in sim-to-real transfer, a good base policy learned from the simulation can be combined with limited real-world data to achieve success. However, effectively utilizing human correction data to address the sim-to-real gap is challenging, especially when we want to prevent catastrophic forgetting of the base policy.

**TRANSIC is effective in addressing different sim-to-real gaps ($\mathcal{Q}2$).** We shed light on its ability to close each individual sim-to-real gap by creating five different simulation-reality pairs. For each of them, we intentionally create large gaps between the simulation and the real world. These gaps are applied to real-world settings, including *perception error*, *underactuated controller*, *embodiment mismatch*, *dynamics difference*, and *object asset mismatch*. Note that these are *artificial* settings for a controlled study. See Appendix C.3 for detailed setups.

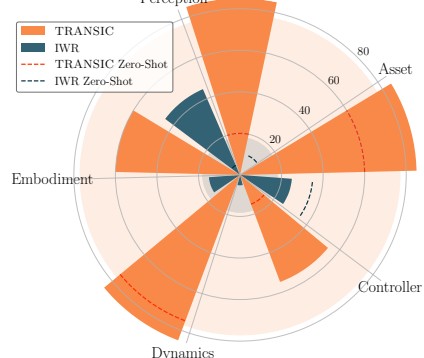

Figure 5: **Robustness to different sim-to-real gaps.** Numbers are averaged success rates (%). Polar bars represent performances after training with data collected specifically to address a particular gap. Dashed lines are zero-shot performances. Shaded circles show average performances.

As shown in Fig. 5, TRANSIC achieves an average success rate of 77% across five different simulation-reality pairs with deliberately exacerbated sim-to-real gaps. This indicates its remarkable ability to close these individual gaps. In contrast, the best baseline method, IWR, only achieves an average success rate of 18%. We attribute this effectiveness in addressing different sim-to-real gaps to the residual policy design. Zeng et al. [80] echos our finding that residual learning is an effective tool to compensate for domain factors that cannot be explicitly modeled. Furthermore, training with data specifically collected from a particular setting generally increases TRANSIC's performance. However, this is not the case for IWR, where fine-

tuning on new data can even lead to worse performance. These results show that TRANSIC is better not only in addressing multiple sim-to-real gaps as a whole but also in handling individual gaps of a very different nature.

**TRANSIC scales with human effort (Q3).** We demonstrate that TRANSIC scales better with human data than the best baseline, IWR, as shown in Fig. 6 and Table A.XII. If we increase the dataset size from 25% to 75% of the full size, TRANSIC improves on average by 42%. In contrast, IWR only achieves a 23% relative improvement. Additionally, for tasks other than *Insert*, IWR performance plateaus at an early stage and even starts to decrease as more human data becomes available. We hypothesize that IWR suffers from catastrophic forgetting and struggles to properly model the behavioral modes of humans and trained robots. On the other hand, TRANSIC bypasses these issues by learning gated residual policies only from human correction.

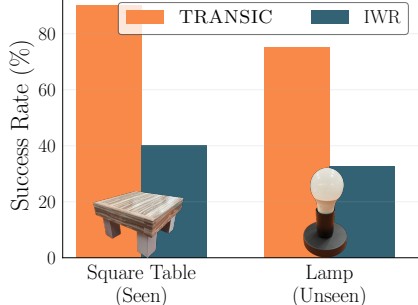

Figure 6: **Scalability with human correction data.** Per-task results are shown in Table A.XII.

**Intriguing properties and emergent behaviors of TRANSIC (Q4).** As shown in Fig. 7, TRANSIC can achieve an average success rate of 75% when zero-shot evaluated on assembling a lamp. However, IWR can only succeed once every three attempts. This evidence suggests that TRANSIC is not overfitting to a particular object; instead, it has learned reusable skills for **category-level object generalization**. Further, with the ablation results shown in Table 1, TRANSIC exhibits intriguing properties including **effective gating**, **policy robustness** against reduced cameras and suboptimal correction data, and **consistency in learned visual features**. See Appendix C.5 for detailed setups and discussions. Qualitatively, TRANSIC shows several rep-

Figure 7: **Generalization to unseen objects from a new category.** Success rates are averaged over tasks *Reach and Grasp* and *Screw*.

resentative behaviors that resemble humans. For instance, they include error recovery, unsticking, safety-aware actions, and failure prevention, as shown in Fig. A.12. Finally, we demonstrate that successful sim-to-real transfer of individual skills can be effectively chained together to enable long-horizon contact-rich manipulation (Fig. 8). See videos at `transic-robot.github.io`.

## 4 Related Work

**Robot Learning via Sim-to-Real Transfer** Physics-based simulations have become a driving force for developing robotic skills [6–10]. However, the domain gap between the simulators and reality is not negligible [40]. Successful sim-to-real transfer includes locomotion [18–

Table 1: **Ablation study results.** Numbers are success rates.

|  | **Average** | Stabilize | Reach and Grasp | Insert | Screw |
|---|---|---|---|---|---|
| Original | 81% | 100% | 95% | 45% | 85% |
| w/ Human Gating | 82% | 100% | 95% | 50% | 85% |
| Reduced Cameras | 80% | 95% | 90% | 40% | 95% |
| Noisy Correction | 76% | 85% | 80% | 45% | 95% |
| w/o Regularization | 55% | 85% | 55% | 20% | 60% |

27], dexterous in-hand manipulation [13–17], and simple non-prehensile manipulation [30–39]. In this work, we tackle more challenging sim-to-real transfer for complex whole-arm manipulation tasks and successfully demonstrate that our approach can solve sophisticated contact-rich tasks. More importantly, it requires significantly fewer real-robot data compared to the behavior cloning approach [68]. This makes solutions based on simulators and sim-to-real transfer more appealing to roboticists.

**Sim-to-Real Gaps in Manipulation Tasks** The sim-to-real gaps can be coarsely categorized as follows: **a)** perception gap [41–43], where synthetic sensory observations differ from those measured in the real world; **b)** embodiment mismatch [18, 44, 45], where the robot models used in simulation do not match the real-world hardware precisely; **c)** controller inaccuracy [46–48], meaning that the results of deploying the same commands differ in simulation and real hardware; and **d)** poor physical realism [49], where physical interactions such as contact and collision are poorly simulated [87]. Traditional methods to address them include system identification [18, 30, 50, 51], domain randomization [13, 52–55], real-world adaptation [56], and simulator augmentation [58–60]. However, system identification is mostly engineered on a case-by-case

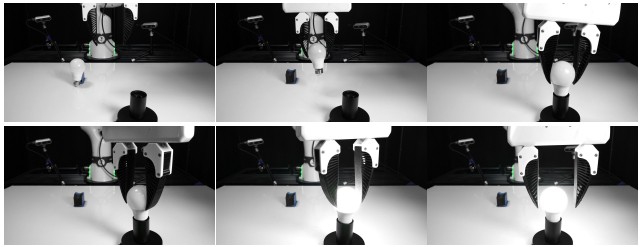

(a) Assemble a lamp (160 seconds in 1× speed).

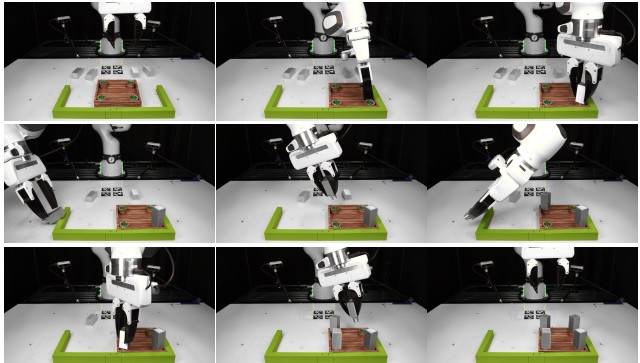

(b) Assemble a square table (550 seconds in 1× speed).

Figure 8: **a)** The robot assembles a lamp. **b)** The robot assembles a square table from FurnitureBench [85]. Videos are available at `transic-robot.github.io`.

basis. Domain randomization suffers from the inability to identify and randomize all physical parameters. Methods with real-world adaptation, usually through meta-learning [88], incur potential safety concerns during the adaptation phase. In contrast, our method leverages human intervention data to implicitly overcome the gap in a domain-agnostic way, leading to a safer deployment.

**Human-in-The-Loop Robot Learning** Human-in-the-loop machine learning is a prevalent framework to inject human knowledge into autonomous systems [62, 89, 90]. The recent trend focuses on continually improving robots' capability with human feedback [91] and autonomously generating corrective intervention data [92]. Our work further extends this trend by showing that sim-to-real gaps can be effectively eliminated by using human intervention and correction signals. In shared autonomy, robots and humans share the control authority to achieve a common goal [64, 65, 93–95]. This control paradigm has been largely studied in assistive robotics and human-robot collaboration [96–98]. In this work, we provide a novel perspective by employing it in the sim-to-real transfer of robot control policies and demonstrating its importance in attaining effective transfer.

## 5 Limitations and Conclusion

**Limitations** 1) Current tasks are in a single-arm tabletop scenario. Though TRANSIC can potentially be applied to more complicated robots with recent teleoperation interfaces [99–103]. 2) Human operators still manually decide when to intervene. This could be automated using failure detection techniques [104, 105]. 3) TRANSIC requires simulation policies with reasonable performance. Nevertheless, it is compatible with recent advances in synthesizing manipulation data [106, 107].

In this work, we present TRANSIC, a human-in-the-loop method for sim-to-real transfer in contact-rich manipulation tasks. We show that combining a strong base policy from simulation with limited real-world data can be effective. However, utilizing human correction data without causing catastrophic forgetting of the base policy is challenging. TRANSIC overcomes this by learning a gated residual policy from a small amount of human correction data. We show that TRANSIC effectively addresses various sim-to-real gaps, both collectively and individually, and scales with human effort.

**Acknowledgments**

We are grateful to Josiah Wong, Chengshu (Eric) Li, Weiyu Liu, Wenlong Huang, Stephen Tian, Sanjana Srivastava, and the SVL PAIR group for their helpful feedback and insightful discussions. This work is in part supported by the Stanford Institute for Human-Centered AI (HAI), ONR MURI N00014-22-1-2740, ONR MURI N00014-21-1-2801, and Schmidt Sciences. Ruohan Zhang is partially supported by the Wu Tsai Human Performance Alliance Fellowship.

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

# A  Simulation Training Details

In this section, we provide details about simulation training, including the used simulator backend, task designs, reinforcement learning (RL) training of teacher policy, and student policy distillation.

## A.1  The Simulator

We use Isaac Gym Preview 4 [9] as the simulator backend. NVIDIA PhysX[1] is used as the physics engine to provide realistic and precise simulation. Simulation settings are listed in Table A.I. The robot model is from Franka ROS package[2]. We borrow furniture models from FurnitureBench [85] to create various tasks that require complex and contact-rich manipulation.



Table A.I: **Simulation settings.**

| Hyperparameter | Value |
|---|---|
| Simulation Frequency | 60 Hz |
| Control Frequency | 60 Hz |
| Num Substeps | 2 |
| Num Position Iterations | 8 |
| Num Velocity Iterations | 1 |



## A.2  Task Implementations

We implement four tasks based on the furniture model `square_table`: *Stabilize*, *Reach and Grasp*, *Insert*, and *Screw*. An overview of simulated tasks is shown in Fig A.1. We elaborate on their initial conditions, success criteria, reward functions, and other necessary information.

### A.2.1  Stabilize

In this task, the robot needs to push the square tabletop to the right corner of the wall such that it is supported and remains stable in following assembly steps. The robot is initialized such that its gripper locates at a neutral position. The tabletop is initialized at the coordinate $(0.54, 0.00)$ relative to the robot base. We then randomly translate it with displacements drawn from $\mathcal{U}(-0.015, 0.015)$ along x and y directions (the distance unit is meter hereafter). We also apply random Z rotation with values drawn from $\mathcal{U}(-15°, 15°)$. Four table legs are initialized in the scene. The task is successful only when the following three conditions are met:

1) The square tabletop contacts the front and right walls;
2) The square tabletop is within a pre-defined region;
3) No table leg is in the pre-defined region.

We use the following reward function:

$$r_t = w_{success}\mathbb{1}_{success} - w_{\dot{\mathbf{q}}}\|\dot{\mathbf{q}}_t\| - w_{action}\|a_t\|, \tag{A.1}$$

where $w_{success}$ is the success reward, $\mathbb{1}_{success}$ indicates the success according to aforementioned conditions, $w_{\dot{\mathbf{q}}}$ penalizes large joint velocities, $\dot{\mathbf{q}}_t$ is the joint velocity, $w_{action}$ penalizes large action commands, and $a_t$ represents the action command at time step $t$. We set $w_{success} = 10$, $w_{\dot{\mathbf{q}}} = 10^{-5}$, and $w_{action} = 10^{-5}$. The episode length is 100. One episode terminates upon success or timeout.

### A.2.2  Reach and Grasp

In this task, the robot needs to reach and grasp a table leg that is randomly spawned in the valid workspace region. The task is successful once the robot grasps the table leg and lifts it for a certain

---

[1]https://developer.nvidia.com/physx-sdk
[2]https://github.com/frankaemika/franka_ros

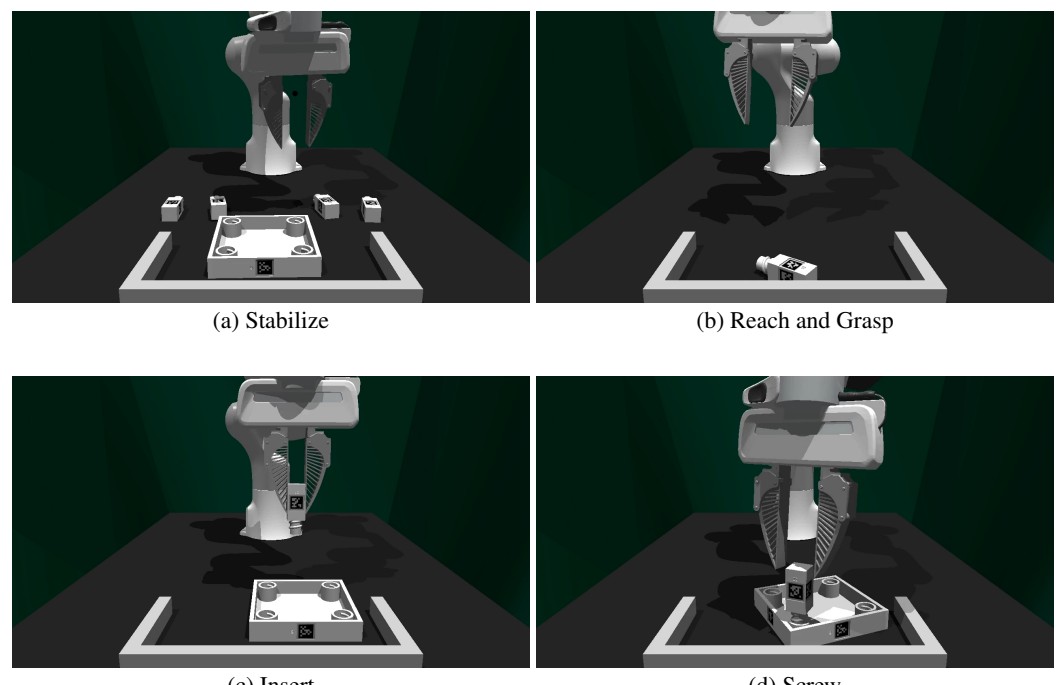

|                    |                    |
|--------------------|--------------------|
| (a) Stabilize      | (b) Reach and Grasp |
| (c) Insert         | (d) Screw          |

Figure A.1: **Visualization of simulated tasks.**

height. The object's irregular shape limits certain grasping poses. For example, the end-effector needs to be near orthogonal to the table leg in the xy plane and far away from the screw thread. Therefore, we design a curriculum over the object geometry to warm up the RL learning. It gradually adjusts the object geometry from a cube, to a cuboid, and finally the table leg. In all curriculum stages, the reward function is

$$r_t = w_{distance}d + w_{lifted}\mathbb{1}_{lifted} + w_{success}\mathbb{1}_{success}. \tag{A.2}$$

Here, $w_{distance}$ is the weight for distance reward, $w_{lifted}$ is the reward for the leg being lifted, and $w_{success}$ is the success weight. $d$ is the distance to the table leg and is calculated as

$$d = 1 - \tanh\left(\frac{10}{4}(d_{eef} + d_{left\_finger} + d_{right\_finger} + d_{orthogonal})\right), \tag{A.3}$$

where $d_{eef}$ is the distance between the end-effector and the table leg, $d_{left\_finger}$ is the distance between the left gripper tip to the table leg, $d_{right\_finger}$ is the distance between the right gripper tip to the table leg, and $d_{orthogonal}$ is the difference between the current and the orthogonal grasping orientations. We set $w_{distance} = 0.1$, $w_{lifted} = 1.0$, and $w_{success} = 200.0$. The episode length is 50. One episode terminates upon success or timeout.

### A.2.3 Insert

In this task, the robot needs to insert a pre-grasped table leg into the far right assembly hole of the tabletop, while the tabletop is already stabilized. The tabletop is initialized at the coordinate $(0.53, 0.05)$ relative to the robot base. We then randomly translate it with displacements sampled from $\mathcal{U}(-0.02, 0.02)$ along x and y directions. We also apply random Z rotation with values drawn from $\mathcal{U}(-45°, 45°)$. We further randomize the robot's pose by adding noises sampled from $\mathcal{U}(-0.25, 0.25)$ to joint positions. The task is successful when the table leg remains vertical and is close to the correct assembly position within a small threshold. We design curricula over the randomization strength to facilitate the learning. The following reward function is used:

$$r_t = w_{distance}d + w_{success}\mathbb{1}_{success}, \tag{A.4}$$

where $w_{distance}$ is the weight for distance-based reward, $d$ is the distance between the table leg and target assembly position, $w_{success}$ is the success weight, and $\mathbb{1}_{success}$ indicates task success. The distance $d$ consists of an Euclidean distance $d_{position}$ and an orientation distance $d_{vertical}$ to encourage the robot to keep the table leg vertical.

$$d = 1 - \tanh\left(\frac{10}{2}\left(d_{position} + d_{vertical}\right)\right) \tag{A.5}$$

We set $w_{distance} = 1.0$ and $w_{success} = 100.0$. The episode length is 100. One episode terminates upon success or timeout.

### A.2.4  Screw

In this task, the robot is initialized such that its end-effector is close to an inserted table leg. It needs to screw the table leg clockwise into the tabletop. We design curricula over the action space: at the early stage, the robot only controls the end-effector's orientation; at the latter stage, it gradually takes full control. We slightly randomize object and robot poses during initialization. The reward function is

$$r_t = (1 - \mathbb{1}_{failure})\left(w_{screw}d_{screw} + w_{success}\mathbb{1}_{success}\right) - w_{deviation}d_{deviation}. \tag{A.6}$$

Here, $\mathbb{1}_{failure}$ indicates the task failure, $w_{screw}$ is the screwing reward weight, $d_{screw}$ measures the screwed angle, $w_{success}$ is the success weight, and $\mathbb{1}_{success}$ indicates the task success. The task is considered as successful when the leg has been screwed $180°$ into the tabletop. It is considered as failed when the table leg tilts more than $10°$ from the vertical pose. We set $w_{screw} = 0.1$, $w_{success} = 100.0$, and $w_{deviation} = 10^{-2}$. The episode length is 200. One episode terminates upon success, failure, or timeout.

## A.3  Teacher Policy Training

### A.3.1  Model Details

**Observation Space**  Besides proprioceptive observations, teacher policies also receive privileged observations to facilitate the learning. They include objects' states (poses and velocities), end-effector's velocity, contact forces, gripper left and right fingers' positions, gripper center position, and joint velocities. Full observations are summarized in Table A.II.

Table A.II: **The observation space for teacher policies.**

| Name | Dimension | | Name | Dimension |
|---|---|---|---|---|
| Proprioceptive | | | Privileged | |
| Joint Position | 7 | | Objects States | $N_{objects} \times 13$ |
| Cosine Joint Position | 7 | | End-Effector Velocity | 6 |
| Sine Joint Position | 7 | | Contact Forces | $N_{objects} \times 3$ |
| End-Effector Position | 3 | | Left and Right Fingers' Positions | 6 |
| End-Effector Rotation | 4 | | Gripper Center Position | 3 |
| Gripper Width | 1 | | Joint Velocity | 7 |

**Controller and Action Space**  An operational space controller (OSC) [73] is used in teacher policy training to improve sample efficiency. We follow Mistry and Righetti [108] to add nullspace control torques to prevent large changes in joint configuration. The action space is thus the change of end-effector's pose. We further add a binary action to control gripper's opening and closing. Formally, it can be expressed as $\mathcal{A}_{teacher} = (\delta x, \delta y, \delta z, \delta r, \delta p, \delta y, \mathbb{1}_{gripper})$, where $(\delta x, \delta y, \delta z) \in \mathbb{R}^3$ is the translation change, $(\delta r, \delta p, \delta y) \in \mathbb{R}^3$ is the rotation change, and $\mathbb{1}_{gripper} \in \{0, 1\}$ is the gripper action.

**Model Architecture**    We use feed-forward policies in RL training. It consists of MLP encoders to encode proprioceptive and privileged vector observations, and unimodal Gaussian distributions as the action head. Model hyperparameters are listed in Table A.III.

Table A.III: **Model hyperparameters for RL teacher policies.**

| Hyperparameter | Value | Hyperparameter | Value |
|---|---|---|---|
| Obs. Encoder Hidden Depth | 1 | Obs. Encoder Activation | ReLU |
| Obs. Encoder Hidden Dim | 256 | Action Head Hidden Layers | [256, 128, 64] |
| Obs. Encoder Output Dim | 256 | Action Head Activation | ELU [109] |

### A.3.2    Domain Randomization

We apply domain randomization during RL training to learn more robust teacher policies. Parameters are summarized in Table A.IV.

Table A.IV: **Domain randomization used in RL training.**

| Parameter | Type | Distribution | Parameter | Type | Distribution |
|---|---|---|---|---|---|
| | **Robot** | | | **Objects** | |
| Mass | Scaling | $\mathcal{U}(0.5, 1.5)$ | Mass | Scaling | $\mathcal{U}(0.5, 1.5)$ |
| Friction | Scaling | $\mathcal{U}(0.7, 1.3)$ | Friction | Scaling | $\mathcal{U}(0.5, 1.5)$ |
| Joint Lower Limit | Scaling | $\log\mathcal{U}(1.00, 1.01)$ | Rolling Friction | Scaling | $\mathcal{U}(0.5, 1.5)$ |
| Joint Upper Limit | Scaling | $\log\mathcal{U}(1.00, 1.01)$ | Torsion Friction | Scaling | $\mathcal{U}(0.5, 1.5)$ |
| Joint Stiffness | Scaling | $\log\mathcal{U}(1.00, 1.01)$ | Restitution | Additive | $\mathcal{U}(0.0, 1.0)$ |
| Joint Damping | Scaling | $\log\mathcal{U}(1.00, 1.01)$ | Compliance | Additive | $\mathcal{U}(0.0, 1.0)$ |

| Parameter | Type | Distribution |
|---|---|---|
| | **Simulation** | |
| Gravity | Additive | $\mathcal{U}(0.0, 0.4)$ |

### A.3.3    RL Training Details

We use the model-free RL algorithm Proximal Policy Optimization (PPO) [81] to learn teacher policies. Hyperparameters are listed in Table A.V. We customize the framework from Makoviichuk and Makoviychuk [110] to use as our training framework.

Table A.V: **Hyperparameters used in PPO training.**

| Hyperparameter | Value | Hyperparameter | Value |
|---|---|---|---|
| Learning Rate | $5 \times 10^{-4}$ | Critic Weight | 4 |
| Discount Factor | 0.99 | GAE [111] $\lambda$ | 0.95 |
| Entropy Weight | 0 | PPO $\epsilon$ | 0.2 |
| Optimizer | Adam [112] | Horizon | 32 |

### A.4    Student Policy Distillation

### A.4.1    Data Generation

We use trained teacher policies as oracles to generate data for student policies training. Concretely, we roll out each teacher policy to generate $10,000$ successful trajectories for each task. We exclude trajectories that are shorter than 20 steps.

### A.4.2 Observation Space

Student policies receive observations that can be obtained in the real world. They are point-cloud and proprioceptive observations. We synthesize point clouds from objects' 6D poses to improve the training throughput. Concretely, given the groundtruth point cloud of the $m$-th object $\mathbf{P}^{(m)} \in \mathbb{R}^{K \times 3}$, we transform it into the global frame through $\mathbf{P}_g^{(m)} = \mathbf{P}^{(m)} \left(\mathbf{R}^{(m)}\right)^\mathsf{T} + \left(\mathbf{p}^{(m)}\right)^\mathsf{T}$. Here $\mathbf{R}^{(m)} \in \mathbb{R}^{3 \times 3}$ and $\mathbf{p}^{(m)} \in \mathbb{R}^{3 \times 1}$ denote the object's orientation and translation in the global frame. Further, the point-cloud representation of a scene $\mathbf{S}$ with $M$ objects is aggregated as $\mathbf{P^S} = \bigcup_{m=1}^{M} \mathbf{P}_g^{(m)}$. For the robot, we only include point clouds for its two fingers and ignore other parts. To facilitate policies to differentiate gripper fingers from the scene, we extend the co-ordinate dimension to include a semantic label $\in \{0, 1\}$ that indicates gripper fingers or not. This information can be obtained on real robots through forward kinematics. A full point cloud is then downsampled to 768 points. Table A.VI lists the observation space.

Table A.VI: **The observation space for student policies.**

| Name | Dimension |
|---|---|
| Point Cloud | $768 \times 4$ |
| Proprioceptive | |
| Joint Position | 7 |
| Cosine Joint Position | 7 |
| Sine Joint Position | 7 |
| End-Effector Position | 3 |
| End-Effector Rotation | 4 |
| Gripper Width | 1 |

### A.4.3 Action Space Distillation

To reduce the controller sim-to-real gap before transfer, we train student policies to output in the configuration space. To achieve that, we relabel actions $\hat{a}$ in trajectories generated by teacher policies from end-effector's delta poses to absolute joint positions. This is equivalent to set $\hat{a}_t = \mathbf{q}_{t+1}$ for all time steps. Therefore, the action space for student policies is $\mathcal{A}_{student} = (\mathbf{q}, \mathbb{1}_{gripper})$, where $\mathbf{q} \in \mathbb{R}^7$ is the joint position within the valid range. In simulation, student policies' actions are deployed with a joint position controller.

### A.4.4 Model Architecture

We use feed-forward policies for tasks *Reach and Grasp* and *Insert* and recurrent policies for tasks *Stabilize* and *Screw* as we find they achieve the best distillation results. PointNets [82] are used to encode point clouds. Recall that each point in the point cloud also contains a semantic label indicating the gripper or not. We concatenate point coordinates with these semantic labels' vector embeddings before passing into the PointNet encoder. We use Gaussian Mixture Models (GMM) [68] as the action head. Detailed model hyperparameters are listed in Table A.VII.

### A.4.5 Data Augmentation

We apply strong data augmentation during distillation. For point-cloud observations, random translation and random jitter are independently applied with a probability $P_{pcd\_aug} = 0.4$. We also add Gaussian noises to proprioceptive observations. Augmentation parameters are listed in Table A.VIII.

### A.4.6 Training Details

To regularize point-cloud features, we separately collect a dataset containing 59 pairs of matched point clouds in simulation and reality. One pair from them is visualized in Fig A.2. Student policies

Table A.VII: **Model hyperparameters for student policies.**

| Hyperparameter | Value | Hyperparameter | Value |
|---|---|---|---|
| Point Cloud | | RNN | |
| PointNet Hidden Dim | 256 | RNN Type | LSTM [113] |
| PointNet Hidden Depth | 2 | RNN Num Layers | 2 |
| PointNet Output Dim | 256 | RNN Hidden Dim | 512 |
| PointNet Activation | GELU [114] | RNN Horizon | 5 |
| Gripper Semantic Embd Dim | 128 | GMM Action Head | |
| Feature Fusion | | Hidden Dim | 128 |
| MLP Hidden Dim | 512 | Hidden Depth | 3 |
| MLP Hidden Depth | 1 | Num Modes | 5 |
| MLP Activation | ReLU | Activation | ReLU |

Table A.VIII: **Data augmentation used in distillation.**

| Hyperparameter | Value | Hyperparameter | Value |
|---|---|---|---|
| Point Cloud | | Proprioception | |
| Augmentation Probability | 0.4 | Prop. Noise Distribution | $\mathcal{N}(0, 0.1)$ |
| Random Translation Distribution | $\mathcal{U}(-0.04, 0.04)$ | Prop. Noise Low | -0.3 |
| Random Jittering Ratio | 0.1 | Prop. Noise High | 0.3 |
| Random Jittering Distribution | $\mathcal{N}(0, 0.01)$ | | |
| Random Jittering Low | -0.015 | | |
| Random Jittering High | 0.015 | | |

are trained by minimizing the loss in Sec. 2.2, where we set $\beta = 10^{-3}$. We use the Adam optimizer [112] with a learning rate of $10^{-4}$ during training. We periodically roll out student policies in simulation for $1,000$ episodes. We then select the checkpoint that corresponds to the highest success rate to use as the base policy in the real-world learning stage.

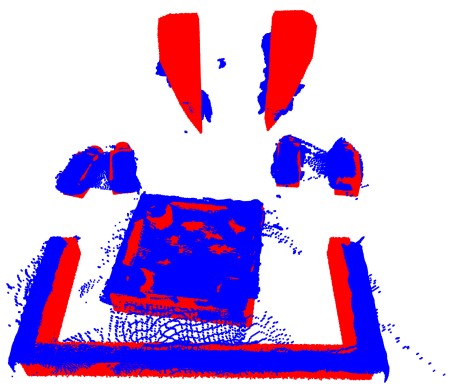

Figure A.2: **Visualization of paired point clouds in simulation (red) and reality (blue).**

## B   Real-World Learning Details

In this section, we provide details about real-world learning, including the hardware setup, human-in-the-loop data collection, and residual policy training.

## B.1 Hardware Setup

As shown in Fig. A.3, our system consists of a Franka Emika 3 robot mounted on the tabletop. We use four fixed cameras and one wrist camera for point cloud reconstruction. They are three RealSense D435 and two RealSense D415. There is also a 3d-printed three-sided wall glued on top of the table to provide external support. We use a joint position controller from the Deoxys library [115] to control our robot at 1000 Hz.

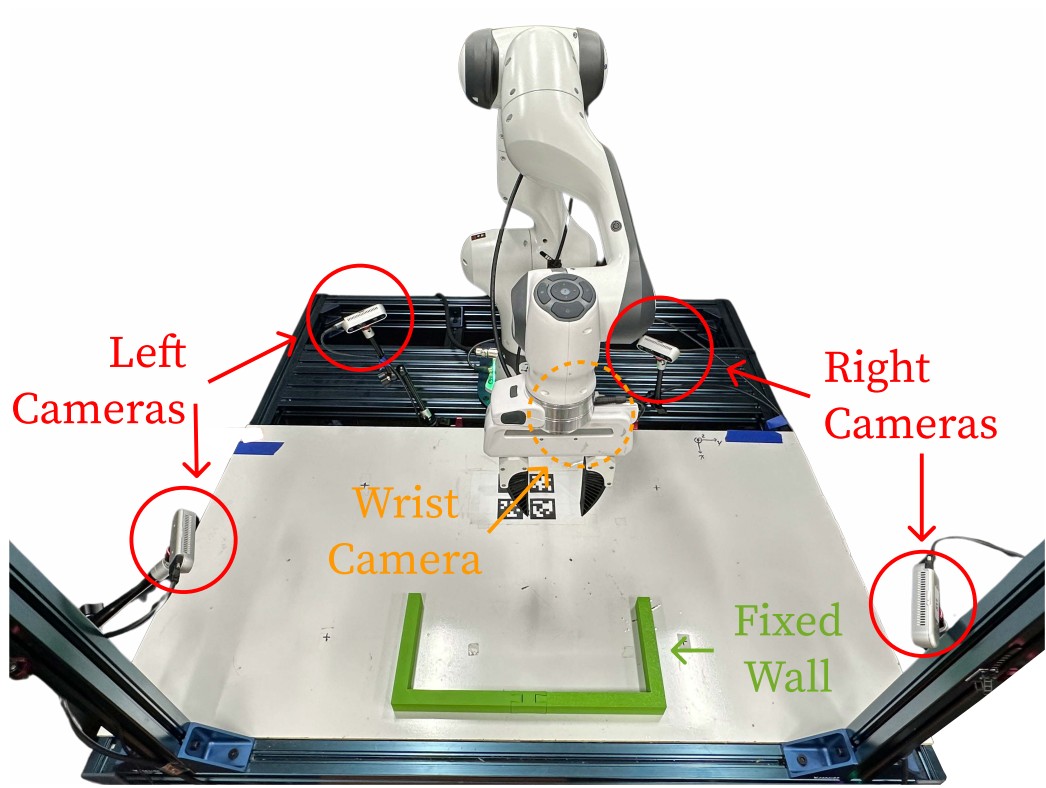

Figure A.3: **System setup.** Our system consists of a Franka Emika 3 robot mounted on the tabletop, four fixed cameras and one wrist camera (positioned at the rear side of the end-effector) for point cloud reconstruction, and a 3d-printed three-sided wall glued onto tabletop to provide external support.

## B.2 Obtaining Point Clouds from Multi-View Cameras

We use multi-view cameras for point cloud reconstruction to avoid occlusions. Specifically, we first calibrate all cameras to obtain their poses in the robot base frame. We then transform captured point clouds in camera frames to the robot base frame and concatenate them together. We further perform cropping based on coordinates and remove statistical and radius outliers. To identify points belonging to the gripper so that we can add gripper semantic labels (Sec. A.4.2), we compute poses for two gripper fingers through forward kinematics. We then remove measured points corresponding to gripper fingers through K-nearest neighbor, given fingers' poses and synthetic point clouds. Subsequently, we add semantic labels to points belonging to the scene and synthetic gripper's point clouds. Finally, we uniformly down-sample without replacement. We opt to not use farthest point sampling [116] due to its slow speed. One example is shown in Fig. A.4.

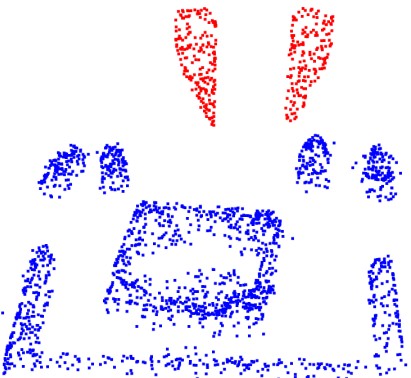

Figure A.4: **Visualization of real-world point-cloud observations.** We obtain them by 1) cropping point clouds fused from multi-view cameras based on coordinates, 2) removing statistical and radius outliers, 3) removing points corresponding to gripper fingers and replacing with synthetic point clouds through forward kinematics, 4) uniformly sampling without replacement, and 5) appending semantic labels to indicate gripper fingers (red) and the scene (blue).

## B.3 Human-in-the-Loop Data Collection

This data collection procedure is illustrated in Algorithm 1. As shown in Fig. A.5, we use a 3Dconnexion SpaceMouse as the teleoperation device. We design a specific UI to facilitate the synchronized data collection. Here, the human operator will be asked to intervene or not. The operator answers through keyboard. If the operator does not intervene, the base policy's next action will be deployed. If the operator decides to intervene, the SpaceMouse is then activated to teleoperate the robot. After the correction, the operator can exit the intervention mode by pressing one button on the SpaceMouse. We use this system and interface to collect 20, 100, 90, and 17 trajectories with correction for tasks *Stabilize*, *Reach and Grasp*, *Insert*, and *Screw*, respectively. We use 90% of them as training data and the remaining as held-out validation sets. We visualize the cumulative distribution function of human correction in Fig. A.6.

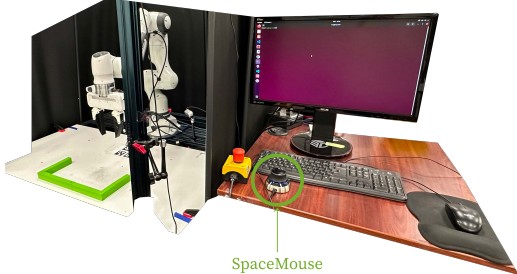

SpaceMouse

Figure A.5: **Real workspace setup for human-in-the-loop data collection.** The human operator provides online correction through a 3Dconnexion SpaceMouse while monitoring the robot's execution.

## B.4 Residual Policy Training

### B.4.1 Model Architecture

The residual policy takes the same observations as the base policy (Table A.VI). Furthermore, to effectively predict residual actions, it is also conditioned on base policy's outputs. Its action head outputs eight-dim vectors, while the first seven dimensions correspond to residual joint positions and the last dimension determines whether to negate base policy's gripper action or not. Besides, a separate intervention head predicts whether the residual action should be applied or not (learned gated residual policy, Sec. 2.4).

**Algorithm 1:** Human Intervention and Online Correction Data Collection

**input** : Base policy $\pi^B$, human policy $\pi^H$, real-world environment $\mathcal{E}$
**output** : Human correction dataset $\mathcal{D}^H$
**initialize:** $\mathcal{D}^H \leftarrow \emptyset$

$o \leftarrow \mathcal{E}.reset()$
**while** *not $\mathcal{E}.terminated$* **do**
    ▷ deploy the base policy
    $a^B \leftarrow a^B \sim \pi^B(o)$
    $o^{next} \leftarrow \mathcal{E}.deploy(a^B)$
    ▷ human decides intervention or not
    $\mathbb{1}^H \leftarrow \pi^H.intervene(o, o^{next})$
    **if** $\mathbb{1}^H$ **then**
        $\mathbf{q}^{pre} \leftarrow \mathcal{E}.robot\_state$
        ▷ deploy human correction
        $a^H \leftarrow a^H \sim \pi^H(o, o^{next})$
        $o^{next} \leftarrow \mathcal{E}.deploy(a^H)$
        $\mathbf{q}^{post} \leftarrow \mathcal{E}.robot\_state$
        ▷ update dataset
        $\mathcal{D}^H \leftarrow \mathcal{D}^H \cup \left(\mathbf{q}^{pre}, \mathbf{q}^{post}, \mathbb{1}^H, o\right)$
    **end**
    ▷ update the next observation
    $o \leftarrow o^{next}$
**end**

For tasks *Stabilize* and *Insert*, we use a PointNet [82] as the point-cloud encoder. For tasks *Reach and Grasp* and *Screw*, we use a Perceiver [83, 84] as the point-cloud encoder. Residual policies are instantiated as feed-forward policies in all tasks. We use GMM as the action head and a simple two-way classifier as the intervention head. Model hyperparameters are summarized in Table A.IX.

Table A.IX: **Model hyperparameters for residual policies.**

| Hyperparameter | Value | Hyperparameter | Value |
|---|---|---|---|
| PointNet | | Feature Fusion | |
| PointNet Hidden Dim | 256 | MLP Hidden Dim | 512 |
| PointNet Hidden Depth | 2 | MLP Hidden Depth | 1 |
| PointNet Output Dim | 256 | MLP Activation | ReLU |
| PointNet Activation | GELU | GMM Action Head | |
| Gripper Semantic Embd Dim | 128 | Hidden Dim | 128 |
| Perceiver | | Hidden Depth | 3 |
| Perceiver Hidden Dim | 256 | Num Modes | 5 |
| Perceiver Number of Heads | 8 | Activation | ReLU |
| Perceiver Number of Queries | 8 | Intervention Head | |
| Gripper Semantic Embd Dim | 128 | Hidden Dim | 128 |
| Base Policy Action Conditioning | | Hidden Depth | 3 |
| Base Policy Gripper Action Embd Dim | 64 | Activation | ReLU |

### B.4.2 Training Details

To train the learned gated residual policy, we first only learn the feature encoder and the action head. We then freeze the entire model and only learn the intervention head. We opt for this two-stage

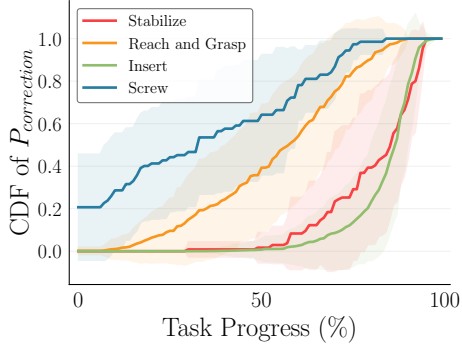

Figure A.6: **Cumulative distribution function (CDF) of human correction.** Shaded regions represent standard deviation. Human correction happens at different times across tasks. This fact necessitates TRANSIC's learned gating mechanism.

training since we find that training both action and intervention heads at the same time will result in sub-optimal residual action prediction. We follow the best practice for policy training, including using learning rate warm-up and cosine annealing [117]. Training hyperparameters are listed in Table A.X.

Table A.X: **Hyperparameters used in residual policy training.**

| Hyperparameter | Value |
|---|---|
| Learning Rate | $10^{-4}$ |
| Weight Decay | $0$ |
| Learning Rate Warm Up Steps | $1,000$ |
| Learning Rate Cosine Decay Steps | $100,000$ |
| Minimal Learning Rate | $10^{-6}$ |
| Optimizer | Adam |

## C    Experiment Settings and Evaluation Details

In this section, we provide details about our experiment settings and evaluation protocols.

### C.1    Task Definition

As shown in Fig. 3, we quantitatively benchmark four tasks. They are fundamental skills required to assemble a square table from FurnitureBench [85]. We randomize objects' initial poses during evaluation.

- *Stabilize*: The robot pushes the square tabletop to the right corner of the wall such that it remains stable in following assembly steps.

- *Reach and Grasp*: The robot reaches and grasps the table leg. It needs to properly adjust the end effector's orientation to avoid infeasible grasping poses.

- *Insert*: The robot inserts the pre-grasped table leg to the far right assembly hole of the tabletop.

- *Screw*: The robot's end-effector is initialized close to an inserted table leg and it screws the table leg clockwise into the tabletop.

## C.2 Main Experiments

We evaluate all methods on four tasks for 20 trials. Each trail starts with different objects and robot poses. We make our best efforts to ensure the same initial settings when evaluating different methods. Specifically, we take pictures for these 20 different initial configurations and refer to them when resetting a new trial. See Figs. A.14, A.15, A.16, A.17 for initial configurations of tasks *Stabilize*, *Reach and Grasp*, *Insert*, and *Screw*, respectively. We follow Liu et al. [91] to label reward for IQL. Full numerical results are provided in Table A.XI.

Table A.XI: **Success rates per tasks.** TRANSIC outperforms all baseline methods in all four tasks.

| Tasks | TRANSIC | Direct Transfer | DR. & Data Aug. [52] | BC Fine-Tune | IQL Fine-Tune | HG-Dagger [66] | IWR [67] | BC [86] | BC-RNN [68] | IQL [69] |
|---|---|---|---|---|---|---|---|---|---|---|
| Stabilize | 100% | 10% | 35% | 55% | 0% | 65% | 65% | 40% | 40% | 5% |
| Reach and Grasp | 95% | 35% | 60% | 35% | 0% | 30% | 40% | 25% | 0% | 5% |
| Insert | 45% | 0% | 15% | 15% | 25% | 35% | 40% | 10% | 5% | 0% |
| Screw | 85% | 0% | 35% | 50% | 65% | 40% | 40% | 15% | 25% | 0% |

## C.3 Experiments with Different Sim-to-Real Gaps

### C.3.1 Experiment Setup

We explain how different sim-to-real gaps are created.

**Perception Error** This is done by applying random jitter to 25% points from point clouds with probability $P = 0.6$, as visualized in Fig. A.7. We test this sim-to-real gap on the task *Reach and Grasp*. The jittering noise is sampled independently from the distribution $\mathcal{N}(0, 0.03)$. We clip the noise to be within the $\pm 0.03$ range.

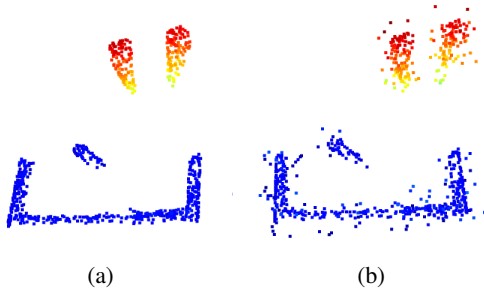

(a) (b)

Figure A.7: **Visualization of introduced perception error. a)** The original point-cloud observation. **b)** The erroneous point-cloud observation with random jitter.

**Underactuated Controller** This is done by making the joint position controller less accurate. We test this gap on the task *Insert*. We emulate an underactuated controller through early stopping. Concretely, at every time a new joint position goal $\mathbf{q}_{goal}$ is set, we record the distance to the goal in configuration space $d_{\mathbf{q}} = \|\mathbf{q} - \mathbf{q}_{goal}\|$ and sample a factor $\Gamma \sim \mathcal{U}(0.80, 0.95)$. The controller will stop reaching the desired goal once it achieves $\Gamma$ progress, i.e., stop early when $\|\mathbf{q} - \mathbf{q}_{goal}\| \leq (1 - \Gamma)d_{\mathbf{q}}$. Fig. A.8 visualizes the effect.

**Embodiment Mismatch** This is done by changing the robot gripper to be shorter length as demonstrated in Fig. A.9. We test this gap on the task *Screw*. We notice that the 9 cm length difference incurs a significant gap.

**Dynamics Difference** This is done by changing object surfaces and increasing friction. We test this gap on the task *Stabilize*. Concretely, we attach friction tapes to the square tabletop's surface to increase friction, hence change the dynamics (Fig. A.10).

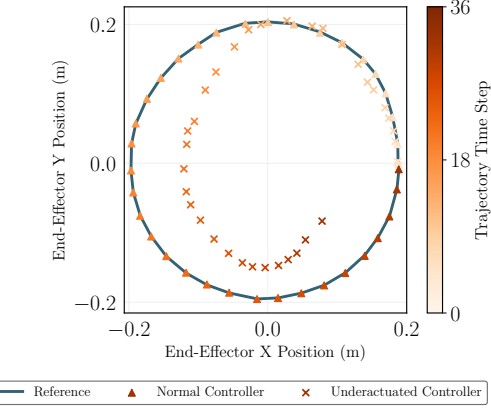

Figure A.8: **Visualization of the trajectory realized by an underactuated controller.** The plot displays the end-effector's position in the XY plane. It shows a reference circular movement, a trajectory tracked by the normal controller, and a trajectory tracked by the underactuated controller.

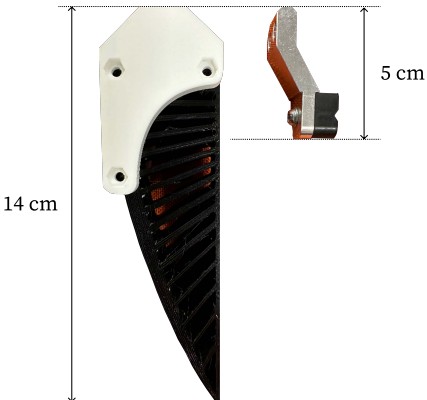

Figure A.9: **Two different gripper fingers used to create embodiment mismatch.** Policies are trained with the longer finger and tested on the shorter finger.

**Object Assert Mismatch** As shown in Fig. A.11, this is done by replacing the table leg with a light bulb. We test this gap on the task *Reach and Grasp*.

### C.3.2 Evaluation

We conduct 20 trails with different initial configurations. Initial conditions for first four experiments are the same as main experiments (Figs. A.14, A.15, A.16, A.17). Fig. A.18 shows initial configurations for the experiment *Object Asset Mismatch*.

### C.4 Data Scalability Experiments

In Table A.XII, we show quantitative results for scalability with human correction dataset size on four tasks.

### C.5 Ablation Studies

### C.5.1 Effects of Different Gating Mechanisms

We introduce the learned gated residual policy in Sec. 2.4 where the gating mechanism controls when to apply residual actions. To assess the quality of learned gating, we compare its performance with an actual human operator performing gating. Results are shown in Table 1 (row "w/ Human

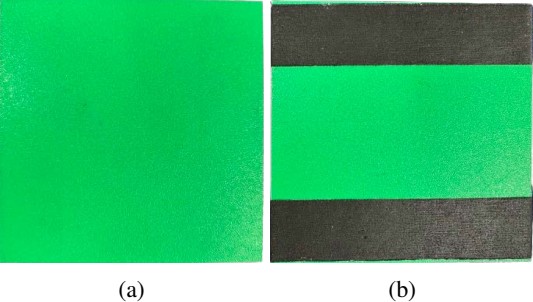

(a)                              (b)

Figure A.10: **Two square tabletops used to create dynamics difference. a)** The original surface is smooth. **b)** We attach friction tapes to change the dynamics.

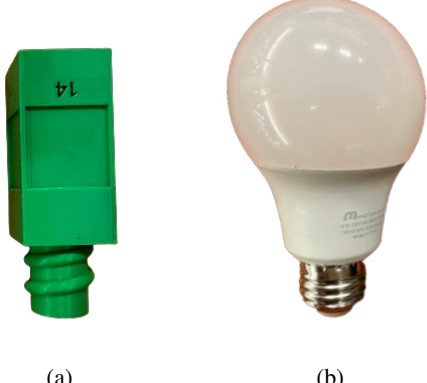

(a)                              (b)

Figure A.11: **Two objects used to create asset mismatch. a)** Policies are trained with the table leg. **b)** We test policies with an unseen light bulb.

Gating"). It is evident that the learned gating mechanism only incurs negligible performance drops compared to human gating. This suggests that TRANSIC can reliably operate in a fully autonomous setting once the gating mechanism is learned.

### C.5.2 Policy Robustness

We investigate the policy robustness against 1) point cloud observations with inferior quality by removing two cameras, and 2) suboptimal correction data with noise injection. We remove two cameras and only keep three. Note that this is the same number of cameras as in FurnitureBench [85]. For tasks other than *Insert*, we keep the wrist camera, the right front camera, and the left rear camera. For the task *Insert*, we keep two front cameras and the left rear camera. We simulate suboptimal correction data by injecting noise into residual actions $a^R$. This noise is of large magnitude, which follows the normal distribution with zero mean and standard deviation corresponding to 5% of the largest residual action in the dataset. Results are shown in Table 1 (rows "Reduced Cameras" and "Noisy Correction"). We highlight that TRANSIC is robust to partial point cloud inputs caused by the reduced number of cameras. We attribute this to the heavy point cloud downsampling employed during training. Fishman et al. [118] echos our finding that policies trained with downsampled synthetic point cloud inputs can generalize to partial point cloud observations obtained in the real world without the need for shape completion. Meanwhile, when the correction data used to learn residual policies are suboptimal, TRANSIC only shows a relative decrease of 6% in the average success rate. We attribute this to the advantage of our integrated deployment—when the residual policy behaves suboptimally, the base policy could still compensate for the error in subsequent steps.

Table A.XII: **Quantitative results for scalability with human correction dataset size on four tasks.**

| Method | Correction Dataset Size (%) | | | | |
|---|---|---|---|---|---|
| | 0 | 25 | 50 | 75 | 100 |
| Stabilize | | | | | |
| TRANSIC | 35% | 80% | 80% | 100% | 100% |
| IWR [67] | | 70% | 75% | 80% | 65% |
| Reach and Grasp | | | | | |
| TRANSIC | 60% | 65% | 80% | 90% | 95% |
| IWR [67] | | 60% | 65% | 40% | 40% |
| Insert | | | | | |
| TRANSIC | 5% | 20% | 35% | 40% | 45% |
| IWR [67] | | 5% | 15% | 30% | 40% |
| Screw | | | | | |
| TRANSIC | 35% | 50% | 65% | 75% | 85% |
| IWR [67] | | 20% | 40% | 40% | 40% |

### C.5.3   Consistency in Learned Visual Features

To learn consistent visual features between the simulation and reality, we propose to regularize the point cloud encoder during the distillation stage. As shown in Table 1 (row "w/o Regularization"), the performance significantly decreases without such regularization, especially for tasks that require fine-grained visual features. Without it, simulation policies would overfit to synthetic point-cloud observations and hence are not ideal for sim-to-real transfer.

### C.6   Qualitative Analysis and Emergent Behaviors

We examine the distribution of the collected human correction dataset. During the human-in-the-loop data collection, the probability of intervening and correcting is reasonably low ($P_{\text{correction}} \approx 0.20$). This is consistent with our intuition that, with a good base policy, interventions are not necessary for most of the time. However, they become critical when the robot tends to behave abnormally due to unaddressed sim-to-real gaps. Moreover, as highlighted in Fig. A.6, interventions happen at different times across tasks. This fact renders heuristics-based methods [119] for deciding when to intervene difficult, and further necessitates our learned residual policy. Several representative behaviors learned by TRANSIC are demonstrated in Fig. A.12.

## D   Additional Experiment Results and Discussions

### D.1   Empirical Justifications for *Action Space Distillation*

Reasons for the proposed *action space distillation* are twofold. The first is mainly because an OSC is hard to sim-to-real transfer, while a joint position controller can be seamlessly transferred. As suggested in Nakanishi et al. [74], an OSC requires accurate modeling of robot parameters, such as the task-space inertia matrix and gravity compensation. System identification helps but is insufficient. Furthermore, it is often the case that given the same joint torque, the end-effector moves differently in simulation and the real world. Because an OSC uses a task-space error to compute joint torques, this will lead to large joint position deviation.

The second is for better training efficiency. As shown in Fig. A.13, it is almost impossible to directly train RL with point-cloud inputs and joint position action space. Even after 7-day training, RL still

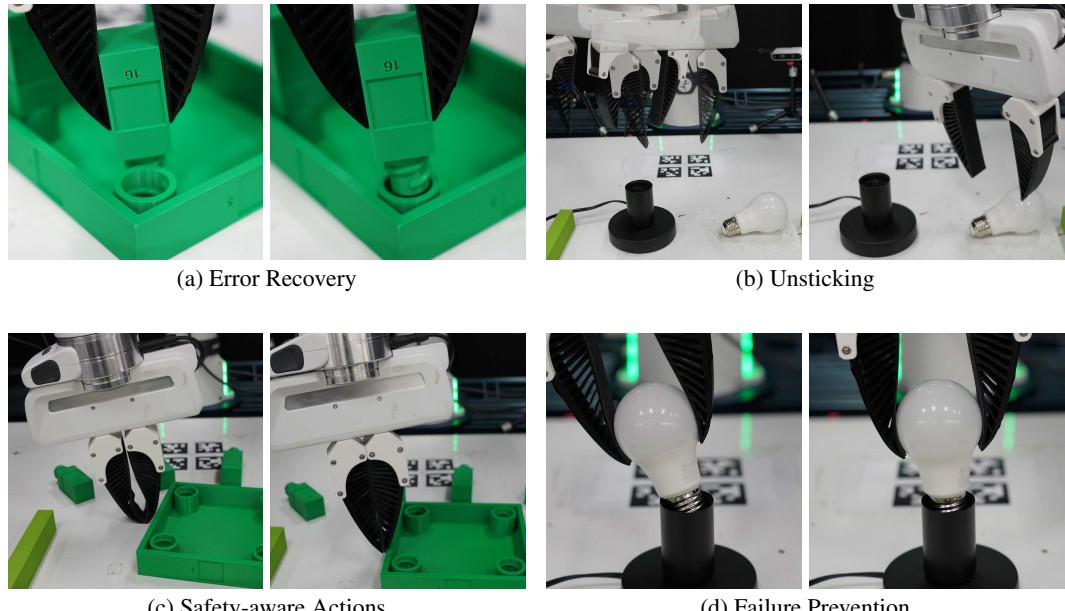

(a) Error Recovery            (b) Unsticking

(c) Safety-aware Actions        (d) Failure Prevention

Figure A.12: **Emergent behaviors learned by TRANSIC. a) Error recovery.** Left: The robot tries to insert the table leg but the direction is wrong; Right: TRANSIC raises the end effector and moves to the correct insertion position. **b) Unsticking.** Left: The robot hovers for a while and never reaches the light bulb; Right: TRANSIC helps the robot get unstuck and move to the bulb. **c) Safety-aware actions.** Left: When pushing the tabletop, the gripper is too low and bends. This might damage the robot; Right: TRANSIC compensates for the command that causes the end effector to move too low. **d) Failure prevention.** Left: The light bulb will fall and break after gripper opening; Right: TRANSIC adjusts the bulb to a stable pose to prevent failure.

shows no sign of improvement. In contrast, TRANSIC takes around 3 days to train on NVIDIA GeForce RTX 3090 GPUs. Therefore, the distillation is important to make the training feasible.

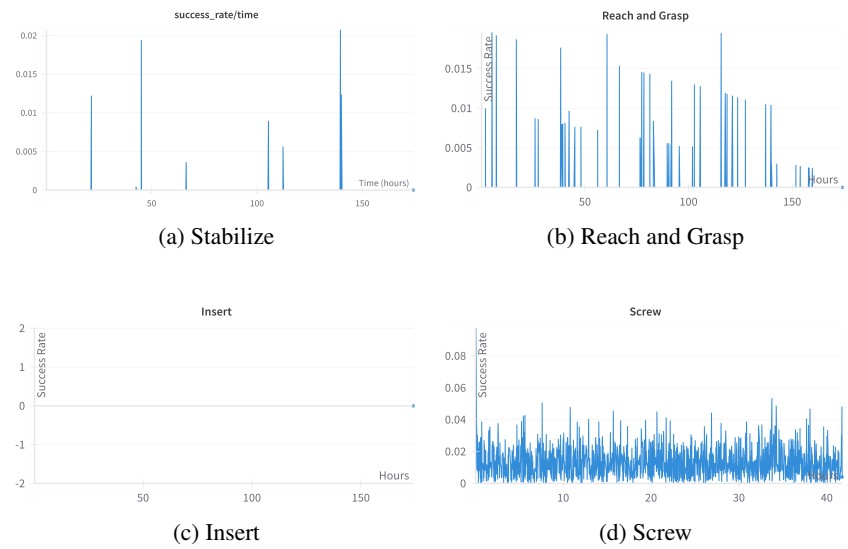

(a) Stabilize             (b) Reach and Grasp

(c) Insert                (d) Screw

Figure A.13: **Learning curves for RL with point-cloud observations and joint position actions.**

### D.2 Distilling Simulation Base Policy with Diffusion Policy

We experiment with learning simulation base policies (Sec. 2.2) with the Diffusion Policy [120]. Concretely, when performing *action space distillation* to learn student policies, we replace the Gaussian Mixture Model (GMM) action head with the Diffusion Policy. Proper data augmentation (Table A.VIII) is also applied to robustify learned policies. Hyperparameters are provided in Table A.XIII.

Table A.XIII: **Diffusion Policy hyperparameters.**

| Hyperparameter | Value | Hyperparameter | Value |
|---|---|---|---|
| Architecture | UNet | $T_o$ | 2 |
| UNet Hidden Dims | [64, 128] | $T_a$ | 8 |
| UNet Kernel Size | 5 | $T_p$ | 16 |
| UNet GroupNorm Num Groups | 8 | Num Denoising Steps (Train) | 100 |
| Diffusion Step Embd Dim | 128 | Num Denoising Steps (Eval) | 16 |

The comparison between GMMs on the real robot is shown in Table. A.XIV. We highlight two findings. First, the significant domain difference between simulation and reality generally exists regardless of different policy parameterizations. Second, since the Diffusion Policy plans and executes a future trajectory, it is more vulnerable to simulation-to-reality gaps due to planning inaccuracy and the consequent compounding error. Only executing the first action from the planned trajectory and re-planning at every step may help, but the inference latency renders the real-time execution infeasible.

Table A.XIV: **The real-robot performance difference between GMM and Diffusion Policy.** The policy error caused by simulation-to-reality gaps will be amplified by the Diffusion Policy because it plans and executes a future trajectory.

|  | **Average** | Stablize | Reach and Grasp | Insert | Screw |
|---|---|---|---|---|---|
| GMM | 33.7% | 35% | 60% | 5% | 35% |
| Diffusion Policy | 22.5% | 35% | 50% | 5% | 0% |

### D.3 Gating Function Justification and Conceptual Comparison

Recall several design choices in the proposed gating mechanism: 1) takes inputs of unstructured sensory observations (point cloud); 2) conditioned on base policy's outputs for effective prediction; 3) the intervention classifier shares the same feature encoder with the residual policy; and 4) the entire pipeline is learned end-to-end. We contrast against several mechanisms from the literature.

Table A.XV: **Gating mechanism conceptual comparison.**

|  | How to decide apply gating or not | Input | Condition on base policy's outputs | Shared feature encoder |
|---|---|---|---|---|
| Ours | End-to-end learned | Point cloud and proprioception | Yes | Yes |
| Residual Policy Learning [79] | No gating | Low-dimensional state | No | No |
| Residual RL [78] | No gating | Low-dimensional state | No | No |
| ThriftyDAgger [119] | Thresholded based on neural network ensemble | Low-dimensional state | No | No |
| Runtime Monitoring [104] | End-to-end learned | RGB and proprioception | No | Yes |

Although this gating function and residual policy can be merged into one, we opt not to do so for better empirical performance. In the task *Screw*, using the gating function with the residual policy achieves a success rate of 85%, while only using the residual policy achieves a success rate of 55%. We hypothesize that this is mainly due to the data imbalance—because intervention and correction happen with a low frequency ($P_{correction} \approx 0.20$), most residual actions are zero. If we naively learn a residual policy from them, it will be biased toward predicting near-zero actions. On the

other hand, by training the residual policy only on human corrections, we offload learning whether to apply the correction to a gating function and thus reduce the negative effects of unbalanced data.

## D.4 Long-Horizon Tasks Statistics

We show statistics about task length from FurnitureBench [85] in Table A.XVI.

Table A.XVI: **Statistics about long-horizon tasks from FurnitureBench [85].**

|  | Number of Steps | Average Human Demo Length |
|---|---|---|
| Lamp | 594 | 2 Minutes |
| Square Table | 1689 | 6 Minutes |

# E  Extended Preliminaries

## E.1  Intervention-Based Policy Learning

We adopt an intervention-based learning framework [66, 67, 91] where a human operator can intervene and take control during the execution of the robot base policy $\pi_B$. Denote the human policy as $\pi_H$, the following combined policy is deployed during data collection:

$$\pi^{deployed} = \mathbb{1}^H \pi^H + \left(1 - \mathbb{1}^H\right) \pi^B, \tag{A.7}$$

where $\mathbb{1}^H$ is a binary function indicating human interventions. Introducing a trajectory distribution $q(\tau)$ that consists of two observation-action distributions generated by the robot $\rho^B$ and human operator $\rho^H$, the original RL objective leads to the maximization of a variational lower bound on logarithmic return [67, 121]:

$$\mathcal{J}(\theta, q) = \mathbb{E}_{q(\tau)} \left[\log R(\tau) + \log p_{\pi_\theta} - \log q(\tau)\right], \tag{A.8}$$

where $p_{\pi_\theta}$ is the induced trajectory distribution. While the human operator optimizes Eq. A.8 through intervention and correction, the robot learner maximizes it through

$$\theta = \arg\max_{\theta \in \Theta} \mathbb{E}_{(s,a)\sim q(\tau)} \left[\log \pi_\theta(a|s)\right]. \tag{A.9}$$

Various intervention-based policy learning methods have been derived by weighting observation-action pairs in Eq. A.9 differently. For example, HG-Dagger [66] completely ignores robot data $\mathcal{D}^B$ and only trains on human data $\mathcal{D}^H$ that contain intervention samples. This is equivalent to $q(\tau) \propto \rho^H$. Intervention Weighted Regression (IWR) [67] balances the data distribution by emphasizing human intervention: $q(\tau) \propto \alpha\rho^H + \rho^B$ with $\alpha = |\mathcal{D}^B|/|\mathcal{D}^H|$. Non-intervention-based methods such as traditional behavior cloning (BC) [86] only learn on $\mathcal{D}^H$ with full human demonstrations instead of intervention. This effectively sets $q(\tau) \propto \rho^H$.

# F  Extended Related Work

***Robot Learning via Sim-to-Real Transfer***  Physics-based simulations [6–10, 49, 122–124] have become a driving force [1, 2] for developing robotic skills in tabletop manipulation [125–128], mobile manipulation [129–132], fluid and deformable object manipulation [133–136], dexterous in-hand manipulation [13–17], locomotion with various robot morphology [18–26, 137], object tossing [80], acrobatic flight [28, 29], etc. However, the domain gap between the simulators and the reality is not negligible [10]. Successful sim-to-real transfer includes locomotion [18–27], in-hand re-orientation for dexterous hands where objects are initially placed near the robot [13–17], and non-prehensile manipulation limited to simple tasks [30–39]. In this work, we tackle more challenging sim-to-real transfer for complex manipulation tasks and successfully demonstrate that our approach can solve sophisticated contact-rich manipulation tasks. More importantly, it requires significantly fewer real-robot data compared to the prevalent imitation learning and offline RL approaches [68, 69, 86]. This makes solutions that are based on simulators and sim-to-real transfer more appealing to roboticists.

***Sim-to-Real Gaps in Manipulation Tasks*** Despite the complex manipulation skills recently learned with RL in simulation [138], directly deploying learned control policies to physical robots often fails. The sim-to-real gaps [10, 40, 44, 139] that contribute to this performance discrepancy can be coarsely categorized as follows: **a)** perception gap [18, 41–43], where synthetic sensory observations differ from those measured in the real world; **b)** embodiment mismatch [18, 44, 45], where the robot models used in simulation do not match the real-world hardware precisely; **c)** controller inaccuracy [46–48], meaning that the results of deploying the same high-level commands (such as in configuration space [140] and task space [141]) differ in simulation and real hardware; and **d)** poor physical realism [49], where physical interactions such as contact and collision are poorly simulated [87].

Although these gaps may not be fully bridged, traditional methods to address them include system identification [18, 30, 50, 51], domain randomization [13, 52–55], real-world adaptation [56], and simulator augmentation [58–60]. However, system identification is mostly engineered on a case-by-case basis. Domain randomization suffers from the inability to identify and randomize all physical parameters. Methods with real-world adaptation, usually through meta-learning [88], incur potential safety concerns during the adaptation phase. Most of these approaches also rely on explicit and domain-specific knowledge about tasks and the simulator *a priori*. For instance, to perform system identification for closing the embodiment gap for a quadruped, Tan et al. [18] disassembles the physical robot and carefully calibrates parameters including size, mass, and inertia. Kim et al. [32] reports that collaborative robots, such as the commonly used Franka Emika robot, have intricate joint friction that is hard to identify and randomized in typical physics simulators. To make a simulator more akin to the real world, Chebotar et al. [39] deploys trained virtual robots multiple times to refine the distributions of simulation parameters. This procedure not only introduces a significant real-world sampling effort, but also incurs potential safety concerns due to deploying suboptimal policies. In contrast, our method leverages human intervention data to implicitly overcome the transferring problem in a domain-agnostic way and also leads to a safer deployment.

***Human-in-The-Loop Robot Learning*** Human-in-the-loop machine learning is a prevalent framework to inject human knowledge into autonomous systems [62, 89, 90]. Various forms of human feedback exist [63], ranging from passive judgement, such as preference [142–151] and evaluation [152–157], to active involvement, including intervention [158–160] and correction [161, 162]. They are widely adopted in solutions for sequential decision-making tasks. For instance, interactive imitation learning [66, 67, 91, 163] leverages human intervention and correction to help naïve imitators address data mismatch and compounding error. In the context of RL, reward functions can be derived to better align agent behaviors with human preferences [145, 148, 149, 152]. Noticeably, recent trend focuses on continually improving robots' capability by iteratively updating and deploying policies with human feedback [91], combining active human involvement with RL [162], and autonomously generating corrective intervention data [92]. Our work further extends this trend by showing that sim-to-real gaps can be effectively eliminated by using human intervention and correction signals.

In shared autonomy, robots and humans share the control authority to achieve a common goal [64, 65, 93–95]. This control paradigm has been largely studied in assistive robotics and human-robot collaboration [96–98]. In this work, we provide a novel perspective by employing it in sim-to-real transfer of robot control policies and demonstrating its importance in attaining effective transfer.

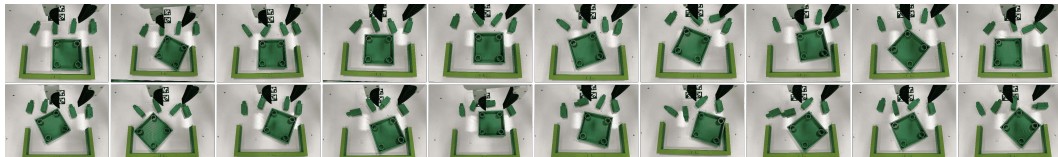

Figure A.14: **Initial settings for evaluating the task *Stabilize*.**

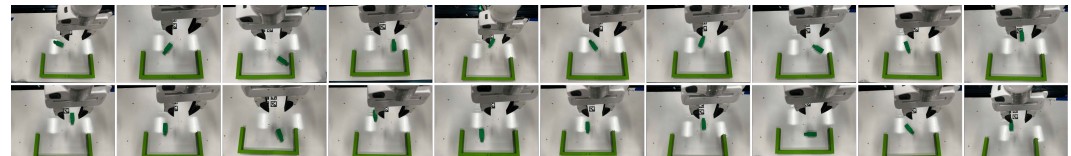

Figure A.15: **Initial settings for evaluating the task *Reach and Grasp*.**

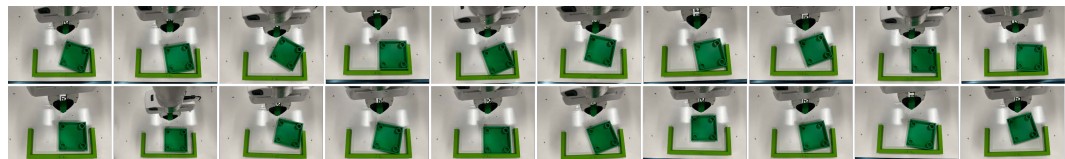

Figure A.16: **Initial settings for evaluating the task *Insert*.**

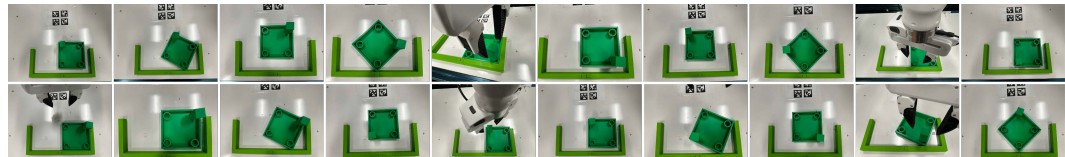

Figure A.17: **Initial settings for evaluating the task *Screw*.**

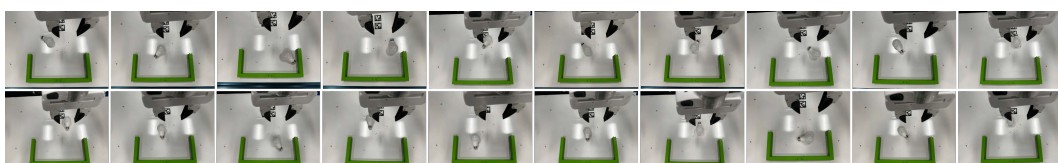

Figure A.18: **Initial settings for the experiment *Object Asset Mismatch*.**

