# OpenReview forum: "TRANSIC: Sim-to-Real Policy Transfer by Learning from Online Correction"
_robot-learning.org/CoRL/2024/Conference — CoRL 2024_

### Official Review · Reviewer_LZdQ · 2024-07-14
**Review for Submission 52**

**Originality:** 2
**Technical Quality:** 3
**Clarity Of Presentation:** 4
**Potential Impact:** 2
**Recommendation:** 3
**Confidence:** 3

**Review:**

This work proposes a human-in-the-loop method for improving sim-to-real transfer of simulation-trained manipulation policies. The advantages of the proposed method are: 1. more effective than traditional sim-to-real methods like domain randomization and data augmentation 2. more data-efficient than imitation learning/offline RL-based methods.

**Strengths:**
1. The problem is well-motivated and the authors not only included a wide variety of tasks but also evaluated different aspects of the sim-to-real gap for robot manipulation.
2. The writing is clear and easy-to-follow. Problems are well-defined and methods are clearly described.
3. The authors compared to a lot of baselines in their evaluations and have done sufficient demonstrations on hardware.

**Weakness:**
1. The overall achieved task success rate is quite low. Without simulation-based policy success rate reported as comparison, it is hard to tell if it is resulting from bad base policy or bad sim-to-real transfer.
2. The number of robot trajectories with human correction seems to differ from task to task, and it is not clear how to determine this number / the upper bound for a new task.
3. It is unclear why a separate gating function is needed. Ideally, the learned residual policy should already incorporate the functionality of the gating function during training (e.g., when correction is not needed, the residual policy predicts close-to-zero action outputs).

**Quality Of The Limitations Section:**

2

**Questions For Rebuttal:**

1. The authors have done a lot of experiments on the real robot. Given the focus of this work is to show effective sim-to-real transfer with the proposed method, what are the success rates of the simulation-trained base policies?

2. In perception error experiments, among all 4 tasks, *insert* and *screw* might be the most sensitive to sensor noise. Would it be possible to show the results on these two tasks?

3. In the limitation section, the authors have mentioned that "Human operators still manually decide when to intervene during correction data collection." After a few trials, can the learned gating function be also used to determine whether human intervention is needed?

4. What are the controller gains used in simulation-based training and real-world deployment? Is there any tuning involved for real-world deployment?

**Robotics Focus:**

4

**Summary Of Paper:**

This work proposes to learn a residual policy from human correction to improve sim-to-real transfer of the simulation-based manipulation policies.

**Summary Of Recommendation:**

Weak reject.

---

### Official Review · Reviewer_pa2D · 2024-07-19
**A novel and effective method to address sim-to-real transfers using online human corrections to train a residual policy**

**Originality:** 3
**Technical Quality:** 4
**Clarity Of Presentation:** 4
**Potential Impact:** 3
**Recommendation:** 3
**Confidence:** 3

**Review:**

### Clarity:
- The paper is well-written, with sufficient details provided in the appendix

### Originality:
- As far as I know, the idea in this paper is novel

### Significance:
- Achieving sim-to-real robustly is a critical problem in robot learning. The paper introduces a very effective method to address the issue in a task-agnostic manner
- The benchmark is nice, challenging enough, and probably will be useful for future research
- The structure of the agent is well-thought, and leverages most state-of-the-art components

### Strengths:
- The set of tasks is challenging, and the method is demonstrated to work well in simulation and hardware, outperforming strong baselines
- The experiments are well organized and very thoughtfully designed. There are great details in the appendix

### Weaknesses:
- Even though the tasks are challenging, they are tabletop and mostly require near top-down actions. Moreover, the existence of the green walls, which can used for alignment, makes the tasks easier to perform

**Quality Of The Limitations Section:**

3

**Questions For Rebuttal:**

- In 2.1, the POMDP formulation is over-killed. The policy is reactive (mapping from the current observation to action), so it is really just an MDP with a high-dimensional state (i.e., point cloud)
- In 2.2, the text description should clarify what kind of privileged observation the teacher is allowed to use
- In Algorithm 1, why the input to \pi^H include both o and o^{next}?
- It is stated in the paper that the residual policy is trained to predict q^{post} – q^{pre}; why such residual policy during deployment can be summed with the base policy to have a working control signal? In other words, it is unclear about the relationship between q^{pre} and the output from the base policy.
- I am curious how easy it is to use the space mouse to generate 6D intervention control inputs.
- I’m surprised that using OSC action space makes sim-to-real transfers more challenging. I thought that high-level action would make transfers easier as we masked out the kinematics, which might differ between sim and real. Can the authors comment on that?
- When compared with baselines, the authors should provide the learning curves (e.g., in the appendix) instead of just providing the final performance. It is useful for future research relying on this approach

**Robotics Focus:**

4

**Summary Of Paper:**

The paper introduces a novel method to tackle sim-to-real problems by introducing humans in the loop to provide correctional action. Human feedback are then used to train a residual policy that will combine with the simulation policy to work well in a set of contact-rich manipulation tasks.

**Summary Of Recommendation:**

The paper presents an effective method to address sim-to-real transfers using human corrections. The approach is novel, and well-supported by experiments.

---

### Official Review · Reviewer_5kGt · 2024-07-20
**Interesting approach that is well-written and rigorously evaluated, with a few approach details that could be clarified**

**Originality:** 4
**Technical Quality:** 5
**Clarity Of Presentation:** 5
**Potential Impact:** 3
**Recommendation:** 4
**Confidence:** 3

**Review:**

# Summary
**Strengths:**
- The approach is conceptually easy to understand and appears to be executed well.
- The experiments are rigorous and convincing, with a robust choice of baselines and demonstration on real robot platforms.
- The paper is excellently written. The literature review appears quite thorough.
- The supplemental videos are very helpful.

**Limitations:**
- It would be beneficial and strengthen the paper to provide additional clarifications to certain aspects of the approach, either in the main paper or in the appendix. (See Approach Clarifications for more notes)
- There are a few writing suggestions for improvements. (See Writing for more notes)

The paper is quite rigorous as written, although the paper can potentially continue to be strengthened through clarifying certain aspects of the approach detailed below.

# Detailed Review

**Approach Clarifications:**
- There isn't much explanation for the dataset D^pcd. It would be helpful to provide more details of this dataset in the main paper or appendix.
- The gating function in Sec 2.4: can you provide more details about this in the main paper? Is this an indicator function? It is not very clear how specifically the gating function works, and this would be helpful to clarify in the main paper as it relates to when/how the residual policy is used.
- According to Sec 2.3, is the human is always involved in the deployment for each timestep? It says "at time step t", but it isn't clear how often this happens. I understand from the limitations that the human operators manually decide when to intervene, but it wasn't clear how often the human is queried for corrections.
- How is the success of a particular task evaluated (e.g., as in Figure 4)?
- Regarding the approach being "domain-agnostic" (line 57 and elsewhere): Although it is agnostic from the robot's perspective, would this not be domain-specific from the human standpoint? If so, the wording "...addressed by humans in a domain-agnostic manner" may be misleading, since it does assume the human has domain-specific knowledge to be able to provide the right interventions to the robot. Perhaps this can be rephrased as saying the interventions are domain-agnostic from the robot's perspective.
- For the lamp assembly task, can you clarify what specific aspect makes it unseen (as stated in Figure 6)? Were point clouds of the lamp objects first collected and used in D^pcd?

**Writing:**
- "Appendix Sec E" -> Could also phrase this (and later references to the Appendix) as Appendix E
- Figure 4b may be better as a separate figure, since it addresses a different question (Q2) than Figure 4a (Q1)
- From my understanding, it seems that IWR is used as the only baseline for Q2, Q3, and Q4 because it is the best baseline (line 224). If so, it would be helpful to put this explanation a bit earlier in the text than when it appears.

**Quality Of The Limitations Section:**

3

**Questions For Rebuttal:**

- Have there been any issues with distribution shifts with point clouds from sim to real? Or, if there were, are such issues correctable with TRANSIC?
- Can you provide more details for D^pcd? Would it be the case that it is required for objects to have real point cloud measurements prior to training the student policy?
- Can you clarify the details of the gating function?
- Can you confirm how often the human is monitoring the executed policy? Is the entire rollout?

**Robotics Focus:**

4

**Summary Of Paper:**

This paper introduces TRANSIC, an approach for sim-to-real policy transfer by learning from online correction and intervention provided by humans. The concept relies on first learning a base policy from simulation, and then learning a residual policy online from human corrections. In simulation, the base policy is trained using a teacher-student framework, where the teacher policy is trained with RL and Operational Space Control, and this teacher policy is then distilled into a student policy that operates on depth clouds with a joint position action space. During the online rollout, a human provides interventions if the robot execution fails. These interventions are stored within a correction dataset and used to train a residual policy. The final policy is synthesized between the base policy and residual policy using a gating function. Experimental results were conducted on real robot platforms and include four tasks relating to furniture assembly and a lamp assembly task.

**Summary Of Recommendation:**

Original Recommendation: As written, the paper has many strong points, such as rigorousness of experimental assessment, quality of writing, and thoroughness of literature review. There are a few aspects of the approach that would be helpful to clarify, either in the main paper or in the appendix. I believe addressing the clarifications can further strengthen the paper, so I would be willing to revise my recommendation based on these clarifications. Updated Recommendation: The author response in the rebuttal has addressed all of my concerns, so I have updated my recommendation to strong accept.

---

### Author Rebuttal · Authors · 2024-08-07

Dear Reviewers and Meta-Reviewer,

We are grateful for the time you have spent providing us with constructive feedback and valuable advice to strengthen our paper. For this rebuttal, we have revised the manuscript and conducted additional experiments to provide more insight and address your concerns. In our responses to each reviewer below, we address your individual questions and comments. The paper has been updated with the suggested revisions, highlighted in yellow. We welcome any follow-up discussions!

---

### Decision · Program_Chairs · 2024-09-04

**Decision:**

Accept

**Comment:**

### Strengths:

- Re: Clarity: The approach is conceptually easy to understand and well-executed (R-5kGt, R-pa2D); R-pa2D additionally mentions that the method is novel and addresses the critical problem of sim-to-real transfer in a task-agnostic manner.
- Re: Experiments: R-5kGt and R-pa2D suggest that the experiments are convincing, with a good choice of baselines, where the method is demonstrated to work well in both simulation and hardware, outperforming strong baselines. R-pa2D and R-LZdQ report that the manuscript considers a wide variety of tasks.
- Re: Writing: all reviewers state that the paper is written clearly, with a good literature review (R-5kGt, R-pa2D) and helpful supplementary material (R-5kGt).
- Re: Methodology: R-LZdQ discussed that the paper's approach is more effective and data-efficient compared to traditional sim-to-real methods like domain randomization and data augmentation.


### Weaknesses:

- Re: Clarity: R-5kGt and R-LZdQ mentioned concerns related to missing details on the dataset DpcdD^{pcd}Dpcd, the gating function, and the specifics of human intervention frequency; also, the description of the POMDP formulation and the privileged observation for the teacher policy needs clarification (R-pa2D).
- Re: Task / Domain: R-pa2D expressed concerns that the tasks are all in tabletop settings and may be easier due to the green walls used for alignment, which could simplify the tasks. R-LZdQ mentioned concerns that the success rates for some tasks are low, making it difficult to assess if the issue lies with the base policy or the sim-to-real transfer.

- Re: Methodology: The need for a separate gating function is questioned, as it should ideally be incorporated into the residual policy (R-LZdQ). R-LZdQ was concerned that the number of robot trajectories with human correction varies across tasks without clear guidelines on determining this number for new tasks.

- Re: Evaluation: The paper lacks detailed comparisons of success rates between simulation-trained base policies and real-world performance (R-LZdQ). More analyses on the sensitivity of tasks to sensor noise and the potential for learned gating functions to automate human intervention decisions are needed (R-LZdQ). The overall impact is seen as incremental rather than significantly advancing the field (R-LZdQ).

### Post-rebuttal Meta Review Statement

I applaud the reviewers and authors for engaging in a productive discussion that resulted in several improvements to the paper, especially as it relates to the clarification of the gating function and addition of new experiments (in response to R-5kGt — enabling them to raise their score), reformulation of the problem (in response to R-pa2D), and the addition of further analysis on variable task complexity (in response to R-LZdQ — enabling them to raise their score).

I recommend for Accept as Poster.

I encourage the authors to follow through with all the promised changes and to incorporate all additional reviewer requests, to improve the paper even further.